**COMMUNICATIONS**

# A lung tropic AAV vector improves survival in a mouse model of surfactant B deficiency

Martin H. Kang [1,2], Laura P. van Lieshout[3,8], Liqun Xu[1,2,8], Jakob M. Domm [3], Arul Vadivel[1,2], Laurent Renesme[1,2], Christian Mühlfeld[4], Maria Hurskainen[1,2], Ivana Mižíková[1,2], Yanlong Pei[3], Jacob P. van Vloten[3], Sylvia P. Thomas[3], Claudia Milazzo[1,2], Chanèle Cyr-Depauw [1,2], Jeffrey A. Whitsett [5], Lawrence M. Nogee [6], Sarah K. Wootton [3,9✉] & Bernard Thébaud[1,2,7,9✉]

Surfactant protein B (SP-B) deficiency is an autosomal recessive disorder that impairs surfactant homeostasis and manifests as lethal respiratory distress. A compelling argument exists for gene therapy to treat this disease, as de novo protein synthesis of SP-B in alveolar type 2 epithelial cells is required for proper surfactant production. Here we report a rationally designed adeno-associated virus (AAV) 6 capsid that demonstrates efficiency in lung epithelial cell transduction based on imaging and flow cytometry analysis. Intratracheal administration of this vector delivering murine or human *proSFTPB* cDNA into SP-B deficient mice restores surfactant homeostasis, prevents lung injury, and improves lung physiology. Untreated SP-B deficient mice develop fatal respiratory distress within two days. Gene therapy results in an improvement in median survival to greater than 200 days. This vector also transduces human lung tissue, demonstrating its potential for clinical translation against this lethal disease.

[1] Sinclair Center for Regenerative Medicine, Ottawa Hospital Research Institute, Ottawa, ON K1Y 4E9, Canada. [2] Department of Cellular and Molecular Medicine, University of Ottawa, Ottawa, ON K1N 6N5, Canada. [3] Department of Pathobiology, Ontario Veterinary College, University of Guelph, Guelph, ON N1G 2W1, Canada. [4] Institute of Functional and Applied Anatomy, Hannover Medical School, 30625 Hannover, Germany. [5] Divisions of Neonatology and Pulmonary Biology, Cincinnati Children's Hospital Medical Center and University of Cincinnati College of Medicine, Cincinnati, OH 45267, USA. [6] Division of Neonatology, Department of Pediatrics, Johns Hopkins University School of Medicine, Baltimore, MD 21205, USA. [7] Neonatology, Department of Pediatrics, Children's Hospital of Eastern Ontario (CHEO) and CHEO Research Institute, Ottawa, ON K1H 8L1, Canada. [8] These authors contributed equally: Laura P. van Lieshout, Liqun Xu. [9] These authors jointly supervised this work: Sarah K. Wootton, Bernard Thébaud. ✉email: kwootton@uoguelph.ca; bthebaud@ohri.ca

Surfactant Protein B (SP-B) is crucial in proper pulmonary surfactant assembly by influencing lipid packing, stabilizing lipid layers, and reducing surface tension[1]. Infants born with SP-B deficiency (OMIM#265120) present with rapidly progressive respiratory failure[2], and without lung transplantation this disease is lethal within 3 to 6 months of birth[3]. Lung transplantation remains a rare procedure in neonates due to preferential allocation of donor lungs for adults or older pediatric patients[4], and unique anatomical and physiological requirements of neonatal patients such as size matching[5]. Genetic causes of pulmonary surfactant protein deficiencies also contribute to an estimated 25% of all severe refractory diffuse lung diseases[6]. This highlights the need to identify alternative therapeutic strategies.

Treatment with exogenous pulmonary surfactant even when enriched for SP-B protein is ineffective[7], and only selective expression of SP-B from alveolar type 2 (AT2) cells maintains proper surfactant homeostasis[8]. Because the lung is a barrier organ in contact with the external environment, targeted delivery of gene therapy to AT2 cells on the alveolar surface has been considered a promising therapeutic strategy. However, earlier attempts at using adenoviral vectors were unsuccessful in expressing SP-B protein in vivo[9,10]. The challenges in delivering transgenes to AT2 cells are formidable due to obstacles such as the immune response, the requirement of cell-surface receptors for entry, and the barrier properties of respiratory mucus and alveolar fluid[11,12]. Alternative therapeutic strategies including *SFTPB* mRNA therapy[13], gene editing[14], and electroporation of *SFTPB* cDNA[15] have been unable to establish a median survival longer than 30 days in SP-B deficient mice[16].

Recently, we engineered an AAV capsid containing an amino acid substitution (F129L) that facilitates heparin binding (AAV6.2) at the cell surface[17], and 2 mutations (Y445F, Y731F) that abrogate ubiquitin-mediated degradation (AAV6.2FF). Although this vector displayed the ability to transduce the lung parenchyma[18], its ability in targeting lung epithelial cells and in vivo therapeutic potential were unexplored.

In this study, we demonstrate that AAV6.2FF transduces airway and alveolar epithelial cells. This rationally designed AAV6 based vector primarily targets cells that demonstrate high expression levels of the cell surface epithelial cell adhesion molecule (EpCAM) marker in the lung parenchyma, which includes AT2 cells. Intratracheal administration of AAV6.2FF delivering either murine or human *SFTPB* cDNA transgene into a SP-B deficient mouse model restores SP-B expression, maintains lamellar body (LB) structure, and improves lung function resulting in extended survival. The clinical relevancy of this gene therapy is demonstrated by the rapid expression of SP-B within days of administration, long-term expression of therapeutic SP-B protein levels, efficacy and safety in neonatal mice, the absence of adverse effects as observed by increases in body weight and the lack of a pro-inflammatory cytokine profile, and the ability to transduce human lung tissue. Due to its propensity to transduce both airway and alveolar epithelial cells, this vector may have the potential to target a number of other monogenetic respiratory diseases.

## Results

**AAV6.2FF transduces airway and alveolar epithelial cells**. To establish the suitability of AAV6.2FF as a vector for lung epithelial cells, the extent and duration of transgene expression in the lungs following intratracheal (IT) administration of $10^{11}$ vector genomes (vg) per mouse of AAV6.2FF-Luciferase (AAV-Luc) was determined. All plasmids contained the inverted terminal repeats from serotype 2 (rAAV2), the CAG or composite CASI promoter validated for lung expression[19],

the woodchuck posttranscriptional regulatory element, and a SV40 polyadenylation tail[20] (Fig. 1a). Comparable increases in body weight to untreated mice indicated that exposure to AAV6.2FF vector did not affect general health (Fig. 1b). Signal as measured by radiance from the thorax (lung) region was observed 7 days post-injection, with a peak at 14 days, followed by a slightly lower but sustained expression over the ensuing 200 days (Fig. 1c, d). We confirmed that luciferase expression originated from the lungs (yellow arrows) through tomographic reconstructions of the In Vivo Imaging System (IVIS) figures (Fig. 1e), and IVIS imaging of precision cut lung slices (PCLS)[21] following AAV-Luc transduction (Supplementary Fig. 1a–c).

To demonstrate cell specificity of AAV6.2FF, $10^{11}$ vg of AAV6.2FF-emGFP-nlsCre vector was administered into Rosa 26 floxed-LacZ reporter mice (Rosa26-Flox/LacZ; JAX Stock No 003474) by intranasal (IN) delivery (Supplementary Fig. 2a, b). Any cell from this transgenic reporter mouse line transduced with this vector results in Cre recombinase-mediated removal of its loxP sites. Subsequent β-galactosidase (β-gal) expression can be observed by X-gal staining. Three weeks after vector delivery, cells lining the nasal cavity, airways, and distal lung were X-gal positive including AT2 and club (Clara) cells (Fig. 2a, b, Supplementary Fig. 2c). However, IN administration did not result in AAV6.2FF vector transduction of non-respiratory organs and tissues (Supplementary Fig. 2d).

Flow cytometry studies further confirmed that AAV6.2FF targets AT2 or ciliated lung epithelial cells. SP-B deficient mice were administered $10^{11}$ vg per mouse of either AAV-Luc (control) or AAV6.2FF-GFP (AAV-GFP) (Supplementary Fig. 3a, b). AT2 cells were stratified from whole dissociated lung tissues based on the lack of CD45 or CD31 staining combined with high (++) epithelial cellular adhesion molecule (EpCAM) expression[22] (Supplementary Fig. 3c–e). This cell population demonstrated high intracellular granularity ostensibly due to the presence of lamellar bodies (LB) (Supplementary Fig. 3f), reinforcing their identity as AT2 cells. In mice administered AAV-GFP, 21.6% of AT2 or ciliated cells expressed GFP (Fig. 2c), while more than 78% of all the cells in the lungs that stained for GFP were AT2 or ciliated cells (Fig. 2d, Supplementary Data File 1). Transduction of AT2 cells was visually supported by immunofluorescence (IF) staining of alveoli from $10^{11}$ vg of AAV-Luc treated mice, which demonstrated co-localization of Luc with proSP-C expressing AT2 cells (Supplementary Fig. 3g, i). Although AAV6.2FF vector transduces multiple cell types following intramuscular (18), or intravenous administration (Supplementary Fig. 4), exclusive targeting of respiratory cells can be ensured by restricting vector delivery through the nasal passage and trachea.

**Murine *proSftpb* cDNA (AAV-mSPB) attenuates lung injury**. To assess the feasibility of AAV6.2FF delivery of *proSftpb* cDNA as a gene therapy for SP-B deficiency, we utilized an inducible transgenic mouse that expresses SP-B in the presence of doxycycline (dox)[16], as the loss of function of both *Sftpb* alleles results in lethal respiratory distress within 20 min of birth[23]. Following dox removal, SP-B expression ceases and the majority of death from respiratory distress occurred within 2 to 7 days, which is similar to the previously reported estimate of 7.5 days ± 3.5 days (4 to 11 days)[16]. Codon optimized murine *proSftpb* cDNA with a C-terminal myc tag was cloned into the rAAV2 expression plasmid (AAV-mSPB; Fig. 3a). AAV-mSPB transduction of HEK293 cells (Multiplicity of Infection = 20,000 vg per cell)[24], or the transient transfection of the pAAV-mSPB expression plasmid resulted in proSP-B expression at the correct

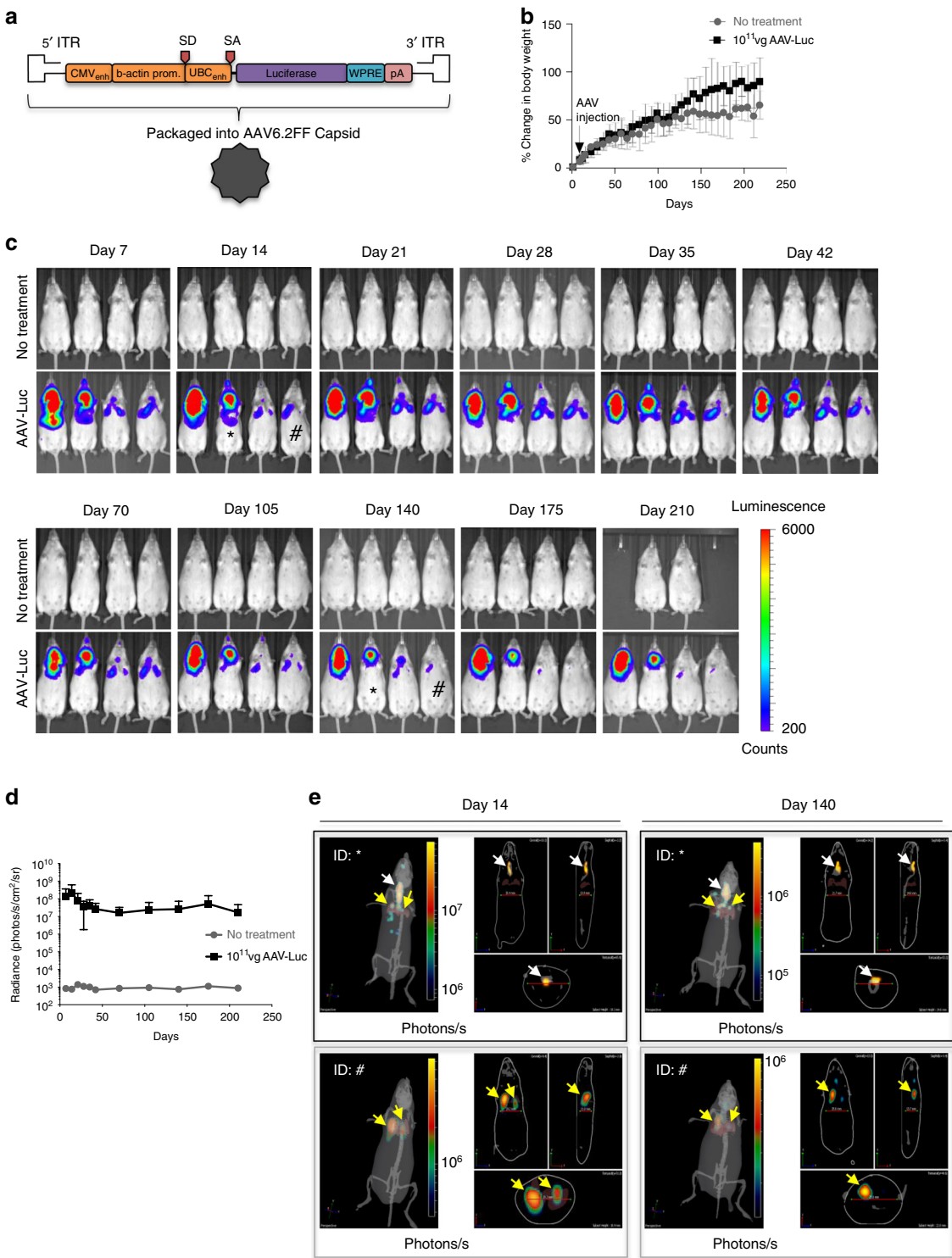

molecular weight (42 kDa)[25] (Fig. 3b). We resolved whether AAV-mSPB prevents the lung phenotype of SP-B deficient mice by comparing mice on dox, to mice administered $10^{11}$ vg per mouse of AAV-Luc or AAV-mSPB off dox (Fig. 4a, Supplementary Fig. 5a). Following IT delivery, we waited 4 weeks before removing the dox, as clinical trials show that transgenes from AAV vectors can take 1 month before reaching peak expression levels[26]. We assessed the lung phenotype 3 days following dox removal, which was before the earliest reported incidence of mortality (4 days)[16].

Body weight revealed itself as a marker of impending respiratory distress. While there were no body weight differences on dox following AAV administration (Fig. 4b, Supplementary Fig. 5b, c), AAV-mSPB mitigated the weight loss observed in AAV-Luc mice following dox removal (Fig. 4c). Three AAV-Luc mice died before the 3-day experimental endpoint and were not included in the assessment of lung function, while all AAV-mSPB mice survived until the harvest on day 3.

Macroscopic tissue injury as indicated by lung lesions (yellow arrows) were observed in 6 out of 7 AAV-Luc mice (Fig. 4d).

**Fig. 1 AAV6.2FF mediates long-term transgene expression in the lungs. a** Schematic of the recombinant AAV2 (rAAV2) vector genome containing the firefly luciferase reporter gene pseudotyped with the AAV6.2FF capsid (AAV-Luc). ITR, inverted terminal repeat; CMV_enh, human cytomegalovirus immediate early gene enhancer region; b-actin prom, chicken beta actin promoter; SD, splice donor; SA, splice acceptor; UBC_enh, human ubiquitin C promoter; WPRE, woodchuck hepatitis virus posttranscriptional regulatory element; pA, simian virus 40 polyadenylation signal. **b** Mean percent change in body weight of untreated mice (gray circles) versus AAV-Luc (black squares) treated mice with standard deviation (SD). $n = 4$ biologically independent animals per group. **c** Bioluminescence detection using the In Vivo Imaging System (IVIS) in transgenic SP-B mice either untreated or intratracheally (IT) administered with $10^{11}$ vector genomes (vg) per mouse of AAV-Luc. Two untreated mice died before imaging on Day 210. All IVIS images were normalized for a signal count ranging from 200 to 6000. The counts scale represents an uncalibrated measurement of photon incidents in a pixel and allows for the normalization between images acquired on different days. **d** Quantification of the IVIS images from the thorax (lung) region in untreated mice (gray circles) versus AAV-Luc (black squares) treated mice. Data are presented as the mean radiance with SD. Radiance is a calibrated measurement of photon emissions and is expressed as the number of photons per second that leave a square centimeter of tissue and radiate into a solid angle of 1 steradian (photons per sec per $cm^2$ per sr; Caliper Life Sciences). ($n = 4$ biologically independent animals per group). **e** Confirmation of lung (yellow arrows) and tracheal (white arrows) region expression by 3D diffuse light imaging tomography (DLIT) reconstruction of the IVIS images on days 14 and 140 in the mice identified by the [*] and [#] symbols. The color scale for the DLIT images are presented as the total flux, which is expressed as the number of photons per second.

Mice on dox showed no visible signs of lung injury, and 3 out of 10 AAV-mSPB mice demonstrated some gross lung damage (yellow arrows). Hematoxylin and Eosin (H&E) and Wright-Giemsa Jenner (WGJ) staining revealed hallmarks of diffuse alveolar damage[27], including infiltration of alveolar spaces by red-blood cells in some AAV-Luc mice (Fig. 4e).

*Sftpb* expression from the vector was confirmed by real-time qPCR (SYBR Green) measurements. We designed the forward primer to bind the murine *proSftpb* sequence, and the reverse primer to the myc sequence in order to differentiate between exogenous transgene and endogenous SP-B expression. An increase greater than 400,000-fold of murine *proSftpb*-Myc was observed in AAV-mSPB treated mice (n = 10) compared to mice on dox (*n* = 10; *P* = 0.0185, one-way ANOVA, Dunnett's post hoc test) in their right lungs (Fig. 4f).

Functionally, AAV-Luc mice appeared to display decreased lung distensibility (Fig. 4g), as well as a decreased %V_10 indicating high alveolar surface tension (Fig. 4h). The %V_10 quantifies the stability of the lung during deflation and decreased values are associated with surfactant inhibition[28–30]. The percentage of the lung volume at 10 $cmH_2O$ divided by the lung volume at 30 $cmH_2O$ yields the %V_10 as demonstrated in Eq. (1).

$$\%V_{10} = V_{10} \div V_{30} \tag{1}$$

In addition, total lung capacity (TLC), residual volume (RV), and compliance[31] were all significantly reduced in AAV-Luc mice compared to mice on dox or in mice treated with AAV-mSPB (Fig. 4i–k). All lung function data was normalized to the body weight of each individual mouse.

IF staining confirmed the presence of proSP-C in all groups; however, SP-B expression was consistent only in mice on dox or following AAV-mSPB treatment. In the AAV-Luc mice, SP-B staining was periodically observed (Supplementary Fig. 5d), suggesting that some endogenous SP-B remained present within 3 days of dox removal. To demonstrate that SP-B expression was solely from the vector, SP-B was stained in the lungs of AAV-mSPB treated mice off dox for 60 days. At the time of harvest these mice showed no signs of respiratory distress, with one mouse displaying only a small region of lung damage (yellow arrow) (Fig. 5a). SP-B staining was comparable to mice on dox, while no SP-B staining was detected in untreated mice off dox after 9 days (Fig. 5b, Supplementary Fig. 5e). All IF-imaging controls are shown in Supplementary Fig. 5f, g.

Pulmonary surfactant is stored and secreted from LB[32,33], the lysosomal-like structures which present with abnormal assembly in SP-B deficient patients[1,2]. Ultrastructural images from AAV-Luc mice revealed small LB often containing central accumulations of amorphous material (white arrows) with few surrounding

lamellae (Fig. 5c, Supplementary Fig. 5h, i). The LB from mice on dox and following AAV-mSPB treatment exhibited the characteristic multi-layered appearance of densely stacked phospholipid membranes[32,33] typical of AT2 cells with proper surfactant homeostasis.

**AAV-mSPB improves survival in a dose-dependent manner.** To determine whether AAV-mSPB improves viability, survival studies were carried out where we monitored for signs of respiratory distress indicating euthanasia (Supplementary Data File 2), or recorded when mice were found expired using the outlined study design (Fig. 6a). All mice were IT administered with a single injection of AAV-mSPB or a negative control (1 × PBS, AAV-Luc, or untreated). Equivalent mortality outcomes using different negative control groups demonstrated the consistency in lethal respiratory distress development in this animal model. A low dose ($10^{10}$ vg per mouse) of AAV-mSPB improved median survival to 5.05 days compared to 3.89 days in 1 × PBS mice (*P* = 0.0067, Log-rank, Mantel-Cox), with the longest surviving mouse living 40 days off dox. At an intermediate dose ($10^{11}$ vg per mouse) of AAV-mSPB, median survival increased to 20 days compared to 3 days in AAV-Luc treated mice (*P* < 0.0001, Log-rank, Mantel-Cox), with 1 mouse surviving for 71 days off dox. In the highest dose ($5 \times 10^{11}$ vg per mouse), we observed a median survival of 128 days (*P* < 0.0001, Log-rank, Mantel-Cox), with 1 mouse surviving for over 6 months (200 days) without dox (Fig. 6b,c). Comparison of the survival curves indicates that AAV-mSPB operates in a dose-dependent manner. The majority (57 out of 102) of negative control animals in our study suffered fatal respiratory distress before the earliest expected endpoint of 4 days. However, some animals (3 out of 102) demonstrated a longer than expected (greater than 11 days) survival, but either showed a considerable decrease in body weight before death (n = 2), (Supplementary Fig. 6a), or exhibited substantial alveolar injury by H&E staining (Supplementary Fig. 6b–f).

**Administration methods for AAV delivery to the distal lung.** An issue for many pre-clinical gene therapy studies in small animal models has been the poor translation of promising results into larger animal models or humans due to a lack of clinical relevancy[34,35], so we explored multiple strategies to address this issue in our study. First, we examined clinically relevant methods for vector distribution in the distal lung. In the clinic, SP-B patients are intubated and placed on a mechanical ventilator as their lungs exhibit a high surface tension often in conjunction with cellular and fluid infiltration[36]. Intubation is one of the most efficient administration routes for gene therapy delivery to the

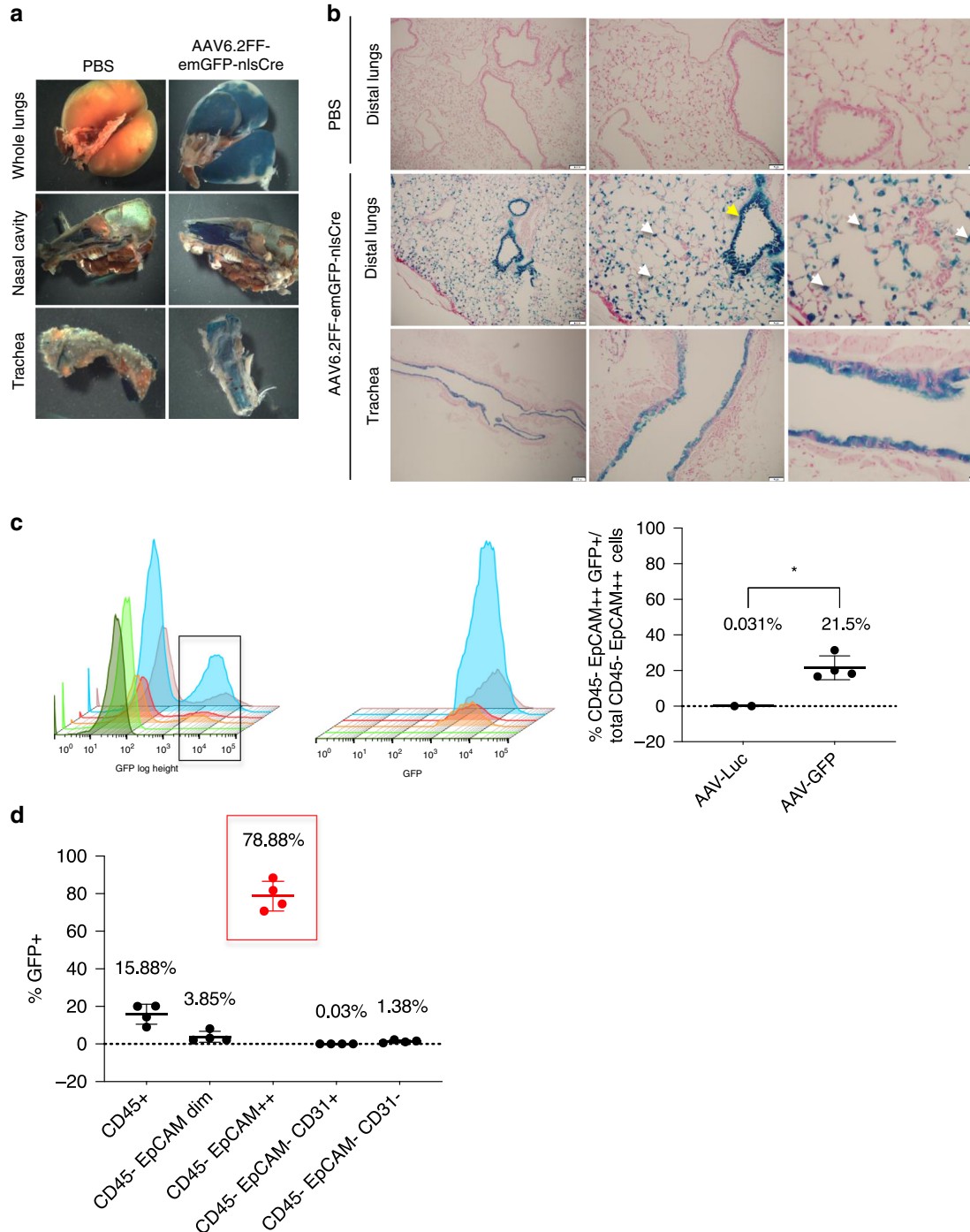

**Fig. 2 AAV6.2FF transduces both airway and alveolar epithelial cells. a** Whole-mount X-gal staining of the lungs, nasal cavity and trachea 3 weeks post-vector administration from either 1 × PBS or AAV6.2FF-emGFP-nlsCre treated Rosa26-Flox/LacZ mice. **b** Representative nuclear fast red counterstained paraffin sections (4 μm) from X-gal stained lungs. LacZ-positive cells were found in the distal airway epithelium and alveoli following delivery of AAV6.2FF-CMV-emGFP-nlsCre. Morphologic criteria demonstrate that both AT2 (white arrows) and club cells (yellow arrow) are X-gal positive (Distal lung scale bars represent 100 μm, 50 μm, and 20 μm from left to right; Trachea scale bars represent 200 μm, 50 μm, and 20 μm from left to right). **c** 3D histograms from flow cytometry analysis of dissociated whole lungs harvested from transgenic SP-B mice 8 days after IT administration of $10^{11}$ vg per mouse of AAV-Luc control ($n = 2$), or $10^{11}$ vg per mouse of AAV6.2FF-GFP (AAV-GFP; $n = 4$). The first set of histograms represent the total CD45 negative, EpCAM++ (high expressing) cell population (first peak), and the CD45 negative, EpCAM++ cells that are GFP positive (second peak) in each mouse. The second set of histograms are the rescaled CD45 negative, EpCAM++, GFP positive cell populations as indicated by the black box in the first histogram. The column graph represents the mean percentage of total CD45 negative, EpCAM++ cells that are GFP positive with SD ($P$ value = two-tailed Student's $t$ test, *$P$ = 0.0127). **d** This column graph represents the mean percentage of different lung cell populations with SD that are GFP positive. CD45 positive staining represents hematopoietic cells, CD45 negative, EpCAM dim staining represents alveolar type 1 (AT1) epithelial cells, CD45 negative, EpCAM++ staining represents AT2 or ciliated cells (red circles and red box), CD45 negative, CD31 positive staining represents endothelial cells, and CD45 negative, EpCAM negative, CD31 negative represents all other cell types found in dissociated lung tissue ($n = 4$ biologically independent animals per group). The source data for **c** and **d** have been provided as a Source Data file.

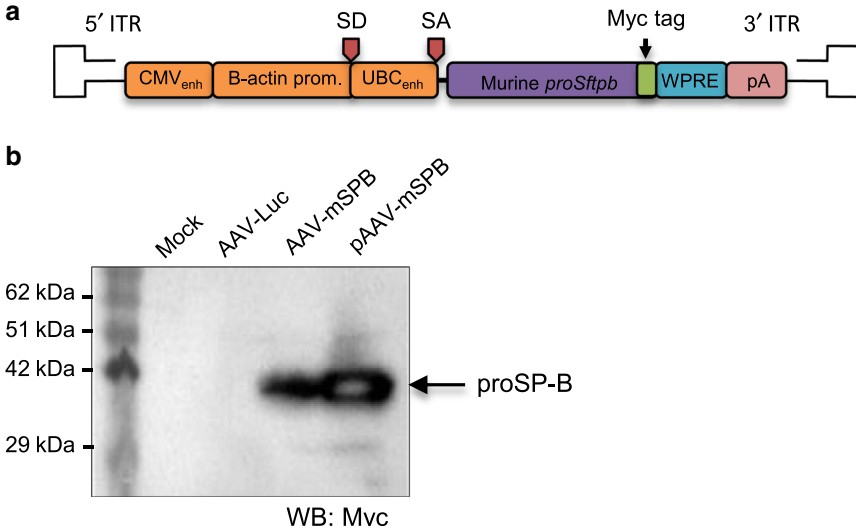

**Fig. 3 Generation of AAV6.2FF murine *proSftpb* cDNA (AAV-mSPB). a** Schematic of codon optimized murine *proSftpb* cDNA transgene in the rAAV2 vector genome. A myc tag is attached to the C-terminal end of the *proSftpb* sequence. **b** Western blot of cell lysates from HEK293 cells transduced with AAV6.2FF-murine *proSftpb* cDNA (AAV-mSPB) at a Multiplicity of Infection (MOI) of 20,000 or transiently transfected with a *proSftpb* expression plasmid (pAAV-mSPB), probed with an anti-myc-tag antibody. The expression of proSP-B protein is indicated by an arrow. The source data for **b** has been provided as a Source Data file.

distal lung[37], and pulmonary surfactant has been effective in promoting adenovirus-mediated gene transfer by facilitating vector dispersion in the distal lung by mucus clearance, and alveolar recruitment through reopening of collapsed or fluid-filled alveoli[38,39]. Both the addition of pulmonary surfactant (Bovine Lipid Extract Surfactant; BLES) as a vehicle and administration through an endotracheal cannula reduced the variability (standard deviation, SD) in the signal of lung reporter expression (Supplementary Fig. 7a–i, Supplementary Data File 3). BLES addition to the IT administration of an intermediate dose ($10^{11}$ vg per mouse) of AAV-mSPB resulted in a median survival of 37 days (Supplementary Fig. 8a, b), and BLES addition to the intubation delivery of an intermediate dose of AAV-mSPB had a median survival of 96 days (Supplementary Fig. 8c, d).

**SP-B rapidly expresses to therapeutic levels**. Second, we determined how soon therapeutic levels of SP-B protein are expressed after AAV administration as rapid SP-B expression from vectors may minimize the alveolar damage caused by surfactant deficiency[36]. SP-B must be expressed above 25% of normal levels in order for proper surfactant homeostasis and respiratory function[16]. We removed dox at 3 (D3) and 7 (D7) days following IT administration of an intermediate dose ($10^{11}$ vg per mouse) of AAV-mSPB. At both time points, enough SP-B protein was present to eliminate the fatalities associated with SP-B deficiency (Fig. 6d, Supplementary Fig. 8e). The median survival for the D3 group was 137 days and for the D7 group 194 days, with one mouse living for 287 days off dox in this latter group.

**AAV-mSPB does not cause an inflammatory cytokine profile**. Third, the safety of AAV-mSPB was assessed by determining whether it caused systemic inflammation in an experiment comparing 3 groups: IT delivery of 1 × PBS maintained on dox ($n = 7$), IT delivery of an intermediate dose of AAV-mSPB with dox removed 7 days after administration ($n = 8$); and intraperitoneal injection of 3 mg per kg of lipopolysaccharides (LPS) from *E. coli* O111:B4 maintained on dox ($n = 8$) (Fig. 6e). LPS is a highly endotoxic component of Gram-negative bacteria and its administration into animals elicits a wide-ranging physiological

response including systemic inflammation. LPS-induced systemic inflammation results in peak inflammatory cytokine levels 2 to 8 h after injection, with a return to baseline within 12 to 24 h[40]. Serum was collected 7 days before (−7D) and 1 (+1D), 7 (+7D), and 29 days (+29D) after AAV or LPS administration. A 3 mg per kg dosage of LPS is considered sublethal in adult mice, however 5 of 8 mice died post-injection, while 1 AAV-mSPB treated mouse died within 5 days following dox removal (Fig. 6f,g, Supplementary Fig. 8f, g). Thirteen humoral factors were analyzed by multiplex technology using the LEGENDplex (BioLegend) mouse inflammation panel. From −7D to +1D, AAV-mSPB administration did not result in a significant increase in any of the 13 cytokines measured (Fig. 6h). Although the effects of LPS on humoral factors was expected to diminish within 24 h, LPS still displayed a significant increase in the inflammatory cytokines IL-23 ($P = 0.020$, Mixed Effects Model, Tukey's post hoc test), TNFα ($P = 0.036$), and GM-CSF ($P = 0.045$) from −7D to +1D.

**Human *proSFTPB* cDNA (AAV-hSPB) also prevents lung damage**. Another approach in improving clinical relevancy was to test whether human *proSFTPB* cDNA (AAV-hSPB) (Fig. 7a) prevents lung injury and improves survival comparable to murine *proSftpb* cDNA (AAV-mSPB) in the SP-B mouse model. A sequence comparison using BLAST demonstrates that there is 69% identity between the murine and human proSP-B amino acid sequence[41]. We compared mice on dox to mice IT administered 1 × PBS + BLES (negative control), or 5 × $10^{10}$ vg AAV-hSPB + BLES. We waited 4 weeks before removing dox, with lung structure and function assessed 4 days following dox removal (Fig. 7b, Supplementary Fig. 9a, Supplementary Data File 4).

Following AAV-hSPB delivery, these mice displayed similar increases in body weight as the other groups while on dox (Fig. 7c; Supplementary Fig. 9b, c), and significantly mitigated the loss in body weight (−1.7%) following dox removal compared to negative control mice (−11.1%; $P = 0.0221$, one-way ANOVA, Tukey's post hoc test) (Fig. 7d). Seven 1 × PBS + BLES mice suffered lethal respiratory distress before the 4-day experimental endpoint and presented with pronounced and extensive lung

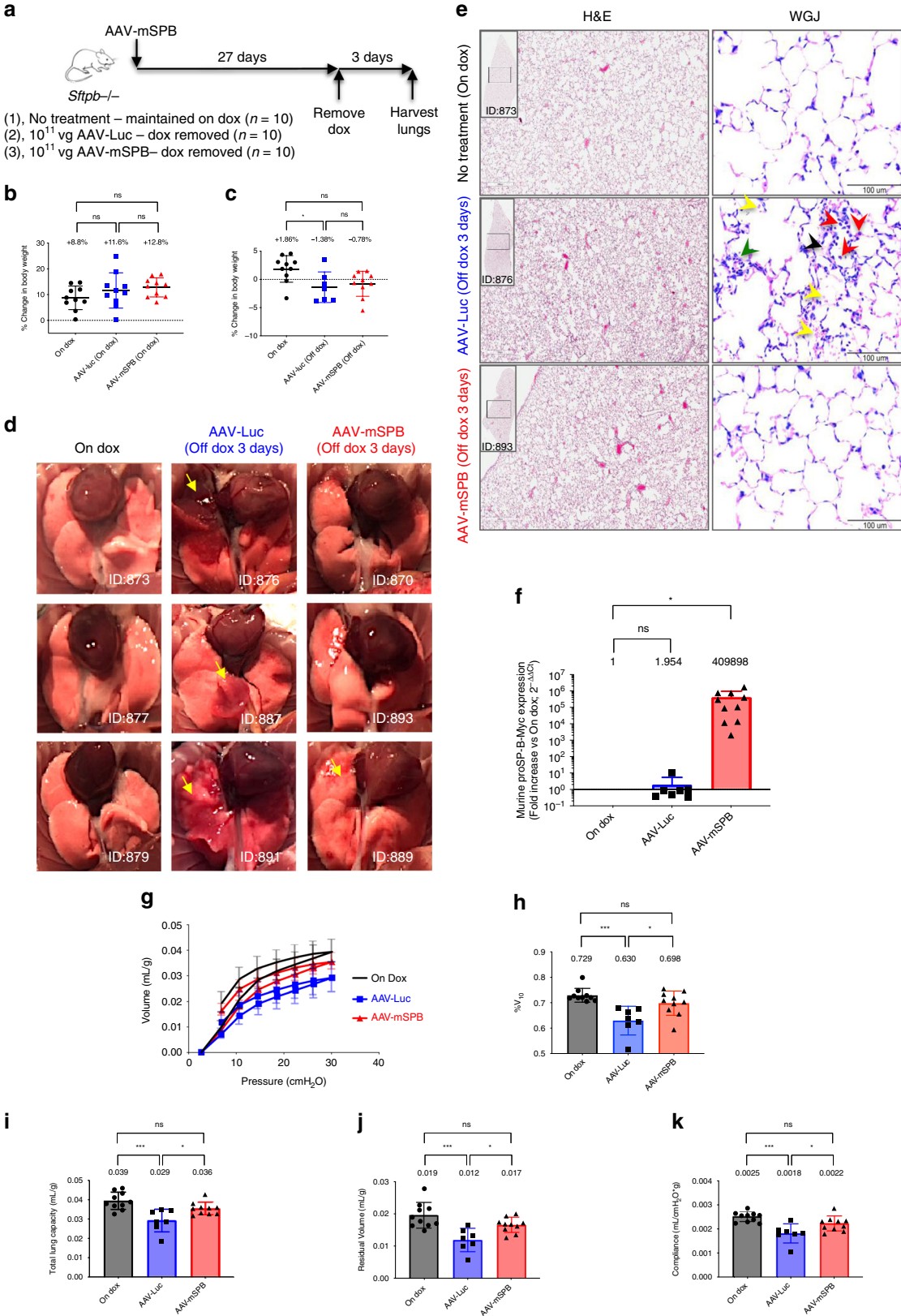

lesions. Macroscopic lung injury (yellow arrows) was also observed in the remaining 1 × PBS + BLES mice that survived, while only 4 out of 11 AAV-hSPB mice demonstrated signs of lung damage (yellow arrows) (Fig. 7e). Representative H&E and WGJ staining revealed large levels of cellular infiltration in the alveolar spaces of 1 × PBS treated negative control mice, but no

signs of alveolar damage in AAV-hSPB treated mice (Fig. 7f, Supplementary Fig. 9d). To demonstrate that lung injury was not caused by post-mortem decomposition, healthy mice on dox euthanized 24 h before lung collection did not exhibit comparable levels of damage as the 1 × PBS treated negative control mice (Supplementary Fig. 9e, f). IF images demonstrated proSP-C

**Fig. 4 AAV-mSPB prevents the lung phenotype of SP-B deficient mice. a** Study design to determine whether $10^{11}$ vg per mouse of AAV-mSPB improves lung structure and function in SP-B deficient mice. **b** Percentage change in body weight following AAV administration while still on doxycycline (dox) (27-day duration). **c** Percentage change in body weight after dox removal (3-day duration). All body weight data in **c** and **d** are presented as the mean with SD ($n = 10$, except AAV-Luc post-dox where $n = 7$; $*P = 0.0294$). **d** Representative gross lung images 3 days following dox removal. Yellow arrows indicate regions of lung injury. Mouse IDs are provided to match with histological or epifluorescence lung images when available. **e** Representative Hemotoxylin and Eosin (H&E) and Wright-Giemsa Jenner (WGJ) staining of paraffin embedded whole left lungs 3 days after dox removal. In the WGJ staining of the AAV-Luc lung, the red arrows indicate the ring-shaped and/or pretzeloid-shaped nuclei of neutrophils, the black arrow indicates hyaline membrane deposition, the green arrow indicates alveolar septal wall thickening, and the yellow arrows indicate red blood cells (H&E scale bar, 500 μm; WGJ scale bar, 100 μm). **f** The expression levels of murine proSP-B-Myc from right lung tissues 3 days after dox removal. All comparative qPCR data are presented as the mean with SD, and the On Dox group acts as a reference point with a value of 1 ($n = 10$, except AAV-Luc where $n = 7$; $*P = 0.0185$). **g** Pressure-volume curve 3 days following dox removal corrected for body weight (in mL per g) for mice on dox (black circles), mice treated with AAV-Luc off dox (blue squares), and mice treated with AAV-mSPB off dox (red triangles). The source data for **g** has been provided as a Source Data file. **h** $\%V_{10}$ corrected for body weight ($*P = 0.0113$, $***P = 0.0003$). **i** Total Lung Capacity (TLC) corrected for body weight (in mL per g; $*P = 0.0243$, $***P = 0.0003$). **j** Residual Volume (RV) corrected for body weight (in mL per g; $*P = 0.0239$, $***P = 0.0003$). **k** Compliance corrected for body weight (in mL per cmH$_2$O per g; $*P = 0.0282$, $***P = 0.0002$). All lung function data are presented as the mean with SD ($n = 10$ except AAV-Luc post-dox where $n = 7$). (All $P$ values in Fig. 4 = 2-tailed ordinary one-way ANOVA with Tukey's or Dunnett's (**f** only) multiple comparisons post hoc test, ns = not significant).

staining in all groups; however, SP-B expression was consistent only in the mice on dox or following AAV-hSPB treatment (Fig. 7g, Supplementary Fig. 9g, h).

Functionally, $1\times$ PBS + BLES mice displayed reduced lung distension (Fig. 7h), as well as surfactant inhibition through a lower $\%V_{10}$ calculated using Eq. (1) compared to AAV-hSPB treated mice (0.571 versus 0.694, respectively; $P = 0.0169$, one-way ANOVA, Tukey's post hoc test) (Fig. 7i). Although TLC, RV, and compliance were all reduced in the $1 \times$ PBS + BLES mice (Supplementary Fig. 9i–k), the differences compared to AAV-hSPB treated mice were not statistically significant due to the reduced sample size of the negative control group ($n = 4$).

**AAV-hSPB improves survival in a dose-dependent manner.** Survival studies following IT administration of $5 \times 10^{10}$ vg per mouse of AAV-hSPB + BLES resulted in a significant improvement in median survival from 3 days ($1 \times$ PBS + BLES) to 11 days ($P = 0.0019$, Log-rank, Mantel-Cox) (Fig. 8a, Supplementary Fig. 9l, Supplementary Movies 1, 2), resembling our findings with the low dose of AAV-mSPB (Fig. 6b). We observed a dose-dependent increase in survival following IT administration of an intermediate dose ($10^{11}$ vg per mouse) of AAV-hSPB + BLES, in which dox was removed 7 days following AAV delivery. In this group, median survival increased to 141 days compared to 6.25 days in $1 \times$ PBS treated mice ($P < 0.0001$, Log-rank, Mantel-Cox), with 1 mouse surviving for 206 days off dox. (Fig. 8a, b, Supplementary Movie 3, Supplementary Data Files 5, 6).

As SP-B deficiency is a neonatal genetic disease, a survival study using neonatal mice was performed to further enhance the clinical relevancy of our gene therapy,. Neonatal transgenic SP-B deficient mice (postnatal (P)8 to P10) were maintained on dox, treated with $1 \times$ PBS + BLES, or treated with $10^{11}$ vg per mouse AAV-hSPB + BLES (Fig. 8c, Supplementary Fig. 9m–o). At P21, the mice were weaned from their mothers and off dox onto a regular chow diet. The $1 \times$ PBS + BLES mice had a median survival of 5 days, while the AAV-hSPB + BLES neonatal mice had a median survival of 205 days with 3 of 7 mice still alive at 230 days at the time of manuscript submission (Fig. 8d, e).

**AAV-hSPB improves survival in models of inherited SP-B deficiency.** In all the preceding experimental designs, AAV-SPB was administered before dox removal. To assess the therapeutic efficacy of AAV-hSPB to treat inherited SP-B deficiency, dox was removed 2 days ($-2$D) and 1 day ($-1$D) before IT administration of an intermediate dose ($10^{11}$ vg per mouse) of AAV-hSPB + BLES (Fig. 9a). A control group with dox removed 1 day before 1

$\times$ PBS + BLES administration ($-1$D) had a median survival of 4 days compared to a median survival of 140 days in the $-1$D group, and 155 days in the $-2$D group following AAV-hSPB + BLES treatment (Fig. 9b). AAV-hSPB restored initial weight loss caused by dox removal, and both treatment groups demonstrated increasing body weight within days of vector administration (Fig. 9c).

To determine whether our vector could treat inherited surfactant deficiency in a neonatal mouse model, P13 to P15 transgenic SP-B pups were removed from dox immediately following IT administration of $10^{11}$ vg per mouse AAV-hSPB + BLES. This was accomplished by placing them with nursing FVB/N (the genetic background of the inducible SP-B model is FVB/N)[16] foster mothers that had recently given birth to similar aged litters (Fig. 9d). Median survival in AAV-hSPB + BLES treated mice was 123 days with 1 mouse still alive at 207 days after dox removal at the time of manuscript submission (Fig. 9e, f, Supplementary Movies 4, 5, Supplementary Data Files 5, 6). The examination of lung histology 3 days after vector administration and dox removal revealed a similar alveolar structure to neonatal mice on dox (H&E staining), with no detectable presence of inflammatory cells such as neutrophils in the alveoli (WGJ staining) (Fig. 9g, h).

**Human lung tissue is transduced by AAV6.2FF.** Finally, we reveal that AAV6.2FF can transduce human fetal lung tissue. PCLS from 2 human fetal lungs (gestational ages $16^{+0}$ and $16^{+3}$ weeks; Fig. 10a, b) were transduced with increasing dosages of AAV-Luc at $10^5$, $10^8$, and $10^{10}$ vg per well, and demonstrated a dose-dependent increase in luciferase expression (Fig. 10c, d, Supplementary Fig. 10a). Transduction of human epithelial lung tissue in developing alveoli was confirmed by co-staining for Luciferase or mCherry with EpCAM in AAV-Luc or AAV-mCherry transduced PCLS, respectively (Supplementary Fig. 10b–e). PCLS from a third pair of human fetal lungs (gestational age $16^{+3}$ weeks; Fig. 10e) were also transduced with $10^{10}$ vg per well of AAV-mSPB or AAV-hSPB. The lung tissues maintained metabolic activity at an equivalent level to untransduced PCLS 3 days after treatment (Fig. 10f), demonstrating the absence of adverse effects on human tissue transduced with AAV6.2FF vector. The transduced human lung tissue slices exhibited an increase in SP-B expression from both vectors by IF staining (Fig. 10g). Although the ages of the fetuses indicate that lung development is in the late pseudoglandular stage[42] and therefore are too early for the development of mature AT2 cells (Supplementary Fig. 10b, d), this finding demonstrates the capability of AAV6.2FF to safely target the human lung parenchyma.

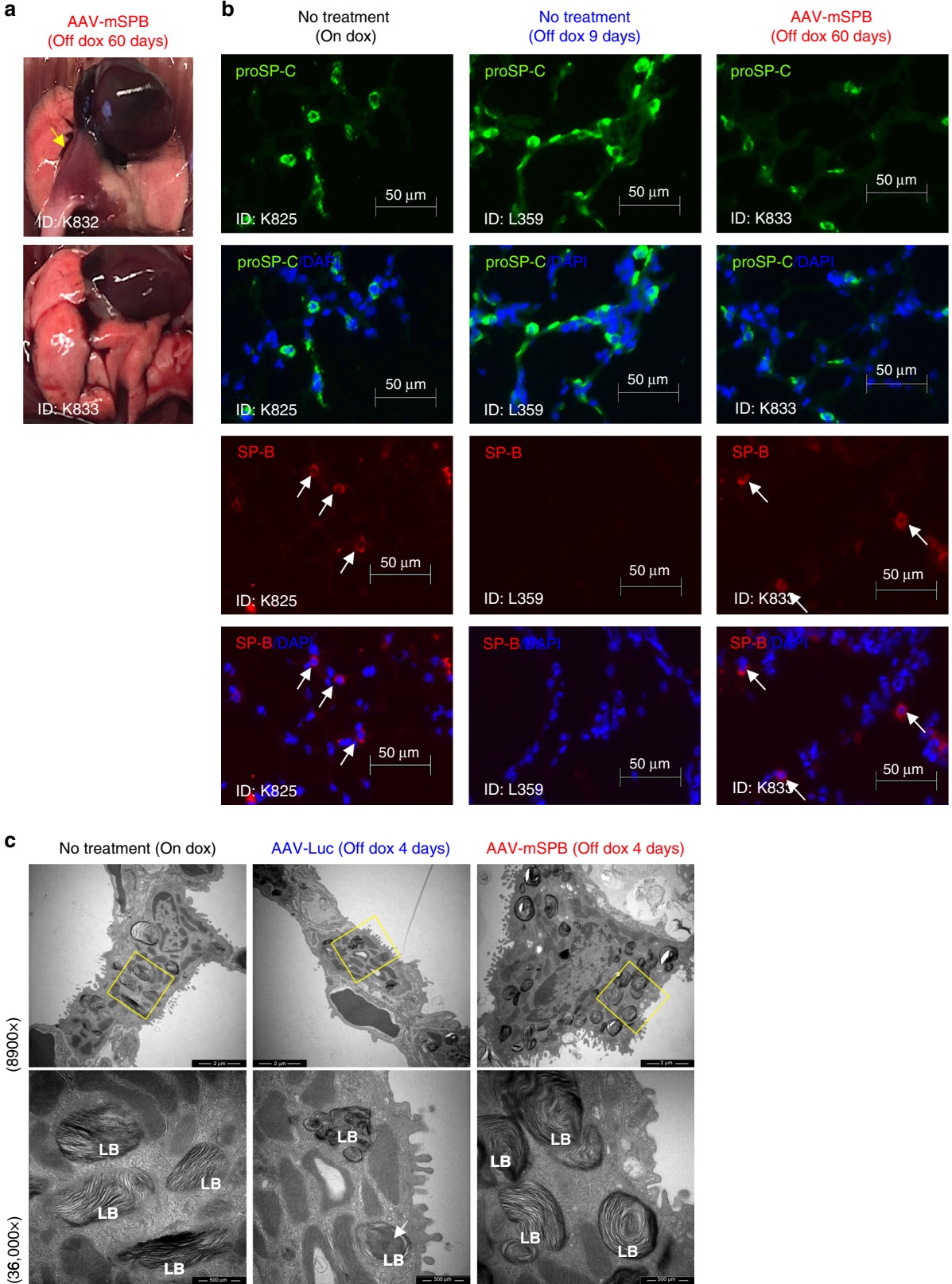

**Fig. 5 AAV-mSPB restores SP-B expression and maintains lamellar body structure. a** Gross lung images from 2 AAV-mSPB treated mice 60 days following dox removal. The yellow arrow indicates a region of lung damage. **b** Representative epifluorescence images of proSP-C (green), SP-B (red), and DAPI (blue) from frozen lung sections of a mouse on dox; an untreated mouse off dox for 9 days; and a $10^{11}$ vg AAV-mSPB treated mouse off dox for 60 days. The white arrows indicate SP-B staining (Scale bar, 50 μm). **c** Representative transmission electron microscope images of AT2 cells 4 days following dox removal. Magnified images (36,000×) of lamellar bodies (LB) are indicated by the yellow boxes on the 8900× images. The white arrow indicates the accumulation of amorphous material within the lamellar body (LB) of an AT2 cell from an AAV-Luc lung (Low magnification image scale bar, 2 μm; High magnification image scale bar, 500 nm).

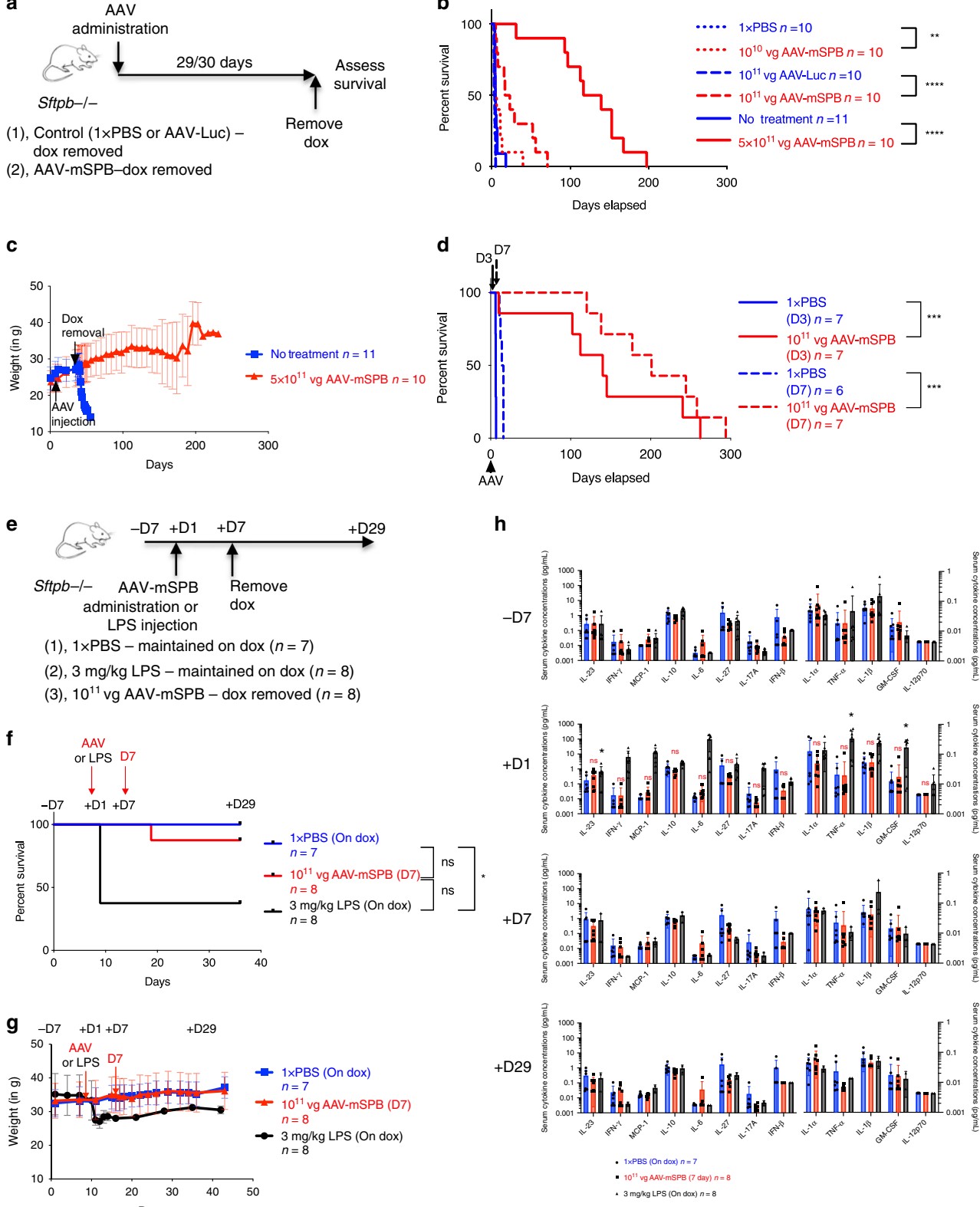

## Discussion

The onset of respiratory impairment in SP-B deficient patients is rapid and lethal[2], and expeditious, long-term SP-B expression is required to feasibly treat this disease. This study demonstrates a safe, rapidly acting, long-term viral-mediated gene therapy with improved survival in a pre-clinical small animal model of SP-B deficiency. We achieved this by overcoming a number of obstacles that confounded earlier attempts at identifying a successful viral-mediated gene therapy strategy for this genetic disorder[9,10].

**Fig. 6 AAV-mSPB improves survival in a dose dependent manner. a** Study design to assess survival following AAV-mSPB gene therapy in SP-B deficient mice. Control mice were either untreated or underwent sham surgeries with IT injections of 1× PBS or AAV-Luc. Mice were maintained on dox following AAV administration for 29 to 30 days. **b** Kaplan-Meier survival curves from IT injections of a low (dotted red line; $10^{10}$ vg per mouse; **$P = 0.0067$), intermediate (dashed red line; $10^{11}$ vg per mouse; ****$P < 0.0001$), and high (solid red line; $5 \times 10^{11}$ vg per mouse; ****$P < 0.0001$) dose of AAV-mSPB. **c** Mean weight measurements with SD of untreated versus mice treated with $5 \times 10^{11}$ vg of AAV-mSPB from **b**. **d** Kaplan-Meier survival curves following dox removal 3 (D3; solid red line; ***$P = 0.0002$) and 7 (D7; dashed red line; ***$P = 0.0005$) days after IT administration of $10^{11}$ vg of AAV-mSPB. **e** Study design to assess inflammation by measuring cytokines from serum in SP-B deficient mice with dox removed 7 days after IT injection of $10^{11}$ vg of AAV-mSPB (red line). The negative control group was administered 1× PBS and maintained on dox (blue line), while a positive control group was administered 3 mg per kg of *E. coli* O111:B4 lipopolysaccharides (LPS) by intraperitoneal (IP) injection (black line). Serum was collected from mice 7 days before (−7D), and 1 (+1D), 7 (+7D), and 29 days (+29D) after AAV or LPS administration. **f** Kaplan-Meier survival curves following dox removal 7 (D7) days after AAV-mSPB or LPS administration (*$P = 0.0133$). The curve is represented until the final day of serum collection (+29D). The complete survival curve is displayed in Supplementary Fig. 8f. **g** Mean weight measurements with SD of all surviving mice from **f**. **h** Measurements of 13 mouse inflammatory cytokines 7 (−7D) days before and 1 (+1D), 7 (+7D), or 29 (+29D) days after AAV or LPS administration. All cytokine measurements are presented as the mean concentration in pg per mL with SD. The source data for **h** has been provided as a Source Data file. (P values in **h** = 2-tailed Mixed Effects Model with the Geisser-Greenhouse correction followed by Tukey's multiple comparisons post hoc test, ns = not significant; IL-23 *$P = 0.0203$; TNF-α *$P = 0.0362$; GM-CSF *$P = 0.0456$) (All survival curve P values in Fig. 6 = 2-tailed Log-rank, Mantel-Cox test, except **f** = 2-tailed Log-rank, Mantel-Cox test with Bonferroni correction, significance is set at $P = 0.0167$, ns = not significant).

A major challenge was to identify a vector capable of efficiently transducing AT2 cells in the distal lung with minimal transduction of other cell types[34]. Using imaging and flow cytometry techniques, AAV6.2FF does transduce AT2 cells, but also other cell types and tissues. Despite this, all the evidence from our pre-clinical experiments using the transgenic SP-B small animal model or human PCLS suggests that AAV6.2FF transduction of multiple cell types is non-toxic. This is based on multiple lines of evidence. The first was body weight measurements. Body weight is a primary indicator of the overall well-being and health of the animal[43], and has been used in previous pre-clinical studies in small animals to demonstrate the safety and effectiveness of gene therapies such as Zolgensma (onasemnogene abeparvovec-xioi)[44]. In our study, AAV6.2FF administration resulted in consistent increases in body weight, although at slightly lower levels than age-matched mice on dox. The second indication for vector safety was the lack of an inflammatory cytokine profile at 1, 7, and 29 days following IT administration of AAV-mSPB. An additional demonstration of vector reliability were the survival studies in which no adverse effects, abnormal behaviors or unexplained injuries were observed in the animals during the course of these experiments. AAV-mSPB mediated long-term survival in these mice with the highest median survival reaching more than 200 days in multiple experiments (Figs. 6d, 8d, Supplementary Fig. 8f), and the longest living subject surviving for 287 days off dox (Fig. 6d). The final piece of evidence supporting vector safety were the resazurin viability assays used to measure metabolic activity in the precision cut lung slices (PCLS) from human embryos. Three days after transduction with AAV-mSPB or AAV-hSPB, tissue slices demonstrated similar levels of viability as untransduced PCLS. From in vivo Cre recombinase studies, we also demonstrate that transduction is localized to the airways or distal lung, with no discernable transduction of non-respiratory tissues, when administration occurs through the trachea or nasal passages.

A potential obstacle in the efficacy of this AAV-mediated gene therapy was our use of single-stranded AAV and the associated delays in transgene expression caused by second strand synthesis[45]. However, survival studies demonstrated that therapeutic levels of SP-B protein were generated within 3 days of AAV-SPB administration, and it is possible that therapeutic levels are generated even more rapidly. In the inherited model of SP-B deficiency experiment (Fig. 9a–c), dox was removed 2 days before AAV administration. With a median survival of 3.65 days in the negative control group, SP-B protein would have been required to express at therapeutic levels within 1 to 2 days of administration

in order to rescue the lethal respiratory distress phenotype. Although our gene therapy shows promising results, the study designs and mouse model still do not faithfully recapitulate the disease progression of human SP-B deficient patients that are born without SP-B protein expression. Even in this experiment where dox was removed before AAV administration (Fig. 9a–c), there was likely endogenous SP-B present when AAV was administered as suggested by SP-B staining in IF images of lungs from AAV-Luc negative control mice 3 days off dox (Supplementary Fig. 5d).

AAV presents a low risk of insertional mutagenesis due to its persistence as episomes, however, long-lasting transgene expression occurs only when non-dividing or self-renewing cell populations are transduced[34]. A subset of AT2 cells are considered to be progenitors, and although the exact rate of AT2 cell turnover is unknown it is estimated to be low[46]. Using a luciferase reporter, we demonstrated stable transgene expression for more than 200 days, while survival studies had a number of mice ($n = 15$ at the time of submission) surviving longer than 200 days (Figs. 6d, 8a, d, Supplementary Fig. 8f).

AAV-SPB acts in a dose-dependent manner with higher dosages demonstrating the most promising therapeutic effects on survival. However, the longest median survivals were observed using an intermediate dose combined with early (day 7 to 13) dox removal (Figs. 6d, 8d, Supplementary Fig. 8f). Part of the improved metrics in survival can be accounted for by the recording of days in which the animals were not on dox while therapeutic expression levels of SP-B were present. This also suggests that the survival studies in which dox was removed 4 weeks after AAV administration markedly underestimates median survival. Eventually the effects of our gene therapy did wane, and we hypothesize that the loss in therapeutic effectiveness is caused by AT2 cell turnover. However, even transient gene therapy can be beneficial in affected individuals. Between 1993 and 2005, 29% of SP-B deficient patients died while waiting for a lung transplant[47], and AAV-SPB therapy represents a practical bridge treatment until surgery can be performed. Re-administration studies potentially combined with steroids or other immunosuppressants are currently being carried out to assess the feasibility of AAV-SPB as a long-term therapy for SP-B deficiency.

AAV6.2FF has applications beyond gene therapy for SP-B deficiency. This vector can provide a therapeutic platform for other genetic disorders of surfactant deficiency affecting AT2 cells including ABCA3 deficiency (OMIM#610921), or SP-C dominant negative mutations (OMIM#610913)[1]. Patients heterozygous for *SFTPB* mutations may also benefit from gene therapy as lung

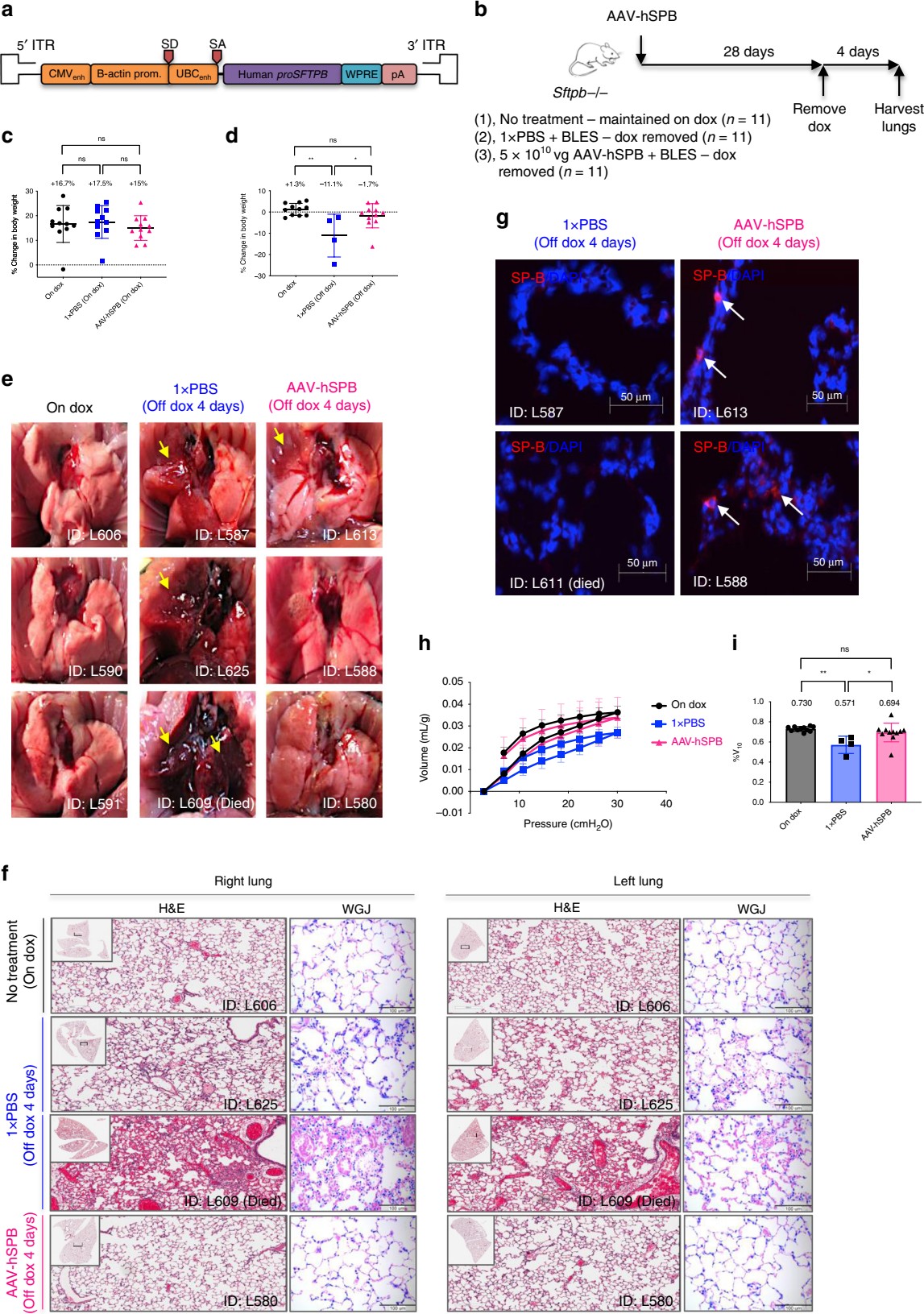

disease can develop when other factors reduce their remaining SP-B levels[36]. An alternative therapeutic strategy for SP-B deficiency is genome editing and AAV6.2FF represents a potential vector for delivery of gene editing constructs into AT2 cells or their progenitors.

In summary, this study demonstrates an effective, long-term viral-mediated gene therapy for genetic SP-B deficiency. Due to the lethal nature of this disease and the lack of therapeutic options, we anticipate rapid clinical translation of this vector.

**Fig. 7 AAV6.2FF human *proSFTPB* cDNA (AAV-hSPB) prevents lung damage. a** Schematic of the human *proSFTPB* cDNA transgene in the rAAV2 vector genome. **b** Study design to determine whether $5 \times 10^{10}$ vg per mouse of AAV-hSPB improves lung structure and function in SP-B deficient mice. **c** Percentage change in body weight following AAV injection while still on dox (duration of 28 days). ($n = 11$ biologically independent animals per group). **d** Percentage change in body weight after dox removal (duration of 4 days). All body weight measurements are presented as the mean with SD ($n = 11$, except $1 \times$ PBS plus BLES post-dox where $n = 4$; $*P = 0.0221$). **e** Representative gross lung images at harvest 4 days following dox removal. Yellow arrows indicate regions of lung injury. The lung image from the mouse found dead was taken within 24 h of death. **f** Representative H&E and WGJ staining of paraffin embedded whole right and left lungs 4 days after dox removal (H&E scale bar, 200 μm; WGJ scale bar, 100 μm). **g** Representative epifluorescence images of SP-B (red) and DAPI (blue) of frozen lung sections from 1× PBS plus BLES treated mice off dox ($n = 2$), and $5 \times 10^{10}$ vg AAV-hSPB plus BLES treated mice off dox ($n = 2$) for 4 days (Scale bar, 50 μm). **h** Pressure-volume curve 4 days following dox removal corrected for body weight (in mL per g) for mice On Dox (black circles), mice treated with 1× PBS plus BLES off dox (blue squares), and mice treated with AAV-hSPB plus BLES off dox (pink triangles). The source data for **h** has been provided as a Source Data file. **i** The %$V_{10}$ corrected for body weight. All lung function data are presented as the mean with SD (For **h** and **i** $n = 11$ except 1× PBS $n = 4$; $*P = 0.0169$, $**P = 0.0021$). (All $P$ values in Fig. 7 = 2-tailed ordinary one-way ANOVA with Tukey's multiple comparisons post hoc test, ns = not significant) (BLES = Bovine Lipid Extract Surfactant).

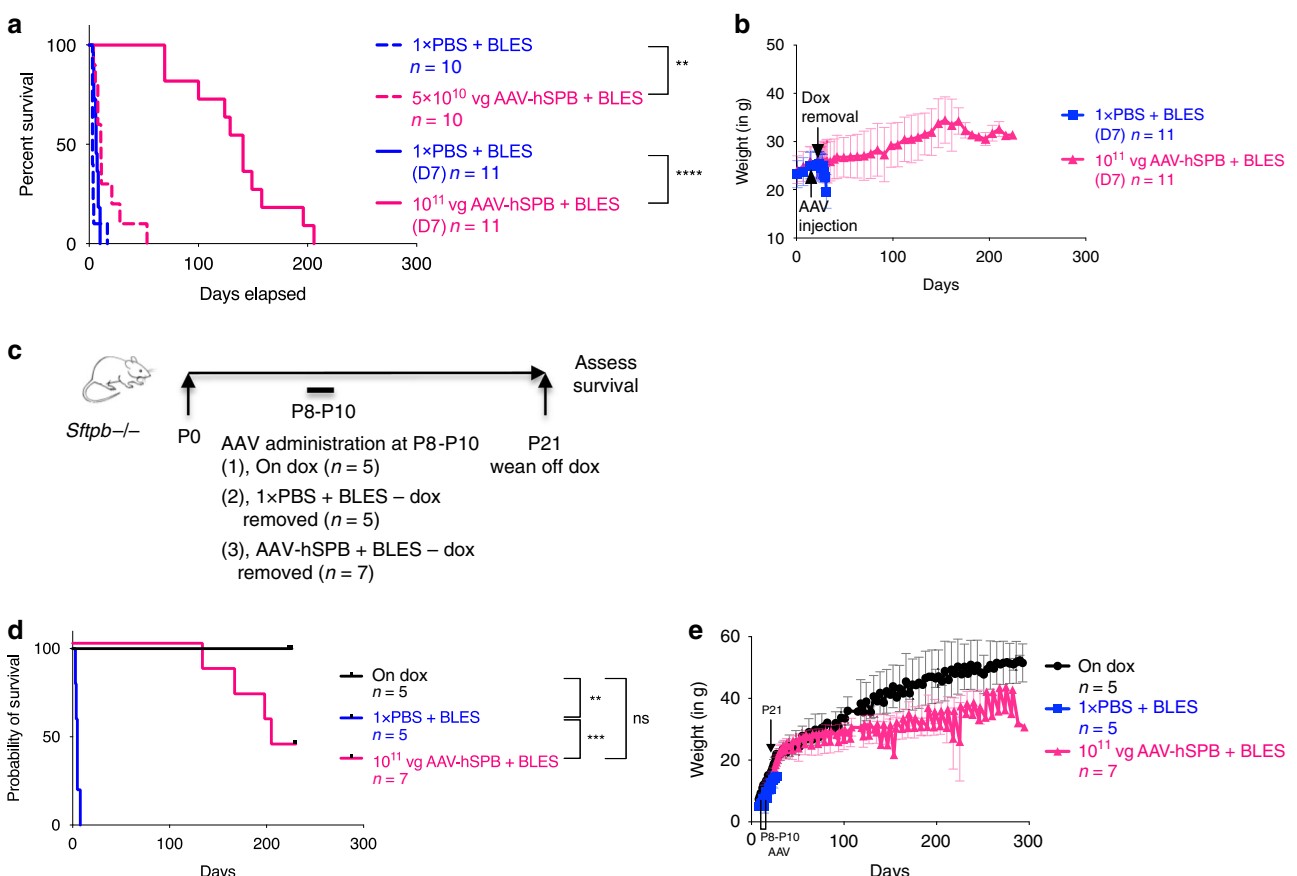

**Fig. 8 AAV-hSPB improves survival in neonatal and adult mice. a** Kaplan-Meier survival curves following IT delivery of $5 \times 10^{10}$ vg per mouse (dashed pink line; $**P = 0.0019$) of AAV-hSPB plus BLES with dox removal 28 days after AAV administration, and IT delivery of $10^{11}$ vg per mouse (solid pink line; $****P < 0.0001$) of AAV-hSPB plus BLES with dox removal 7 days (D7) after AAV administration. **b** Mean weight measurements with SD of all surviving mice either treated with 1× PBS (blue squares) or $10^{11}$ vg of AAV-hSPB plus BLES (pink triangles) from **a**. **c** Study design to assess survival following IT delivery of $10^{11}$ vg per mouse AAV-hSPB plus BLES gene therapy in postnatal (P)8 to P10 SP-B deficient mice. Mice were removed from dox upon weaning at P21. **d** Kaplan-Meier survival curves following IT delivery of $10^{11}$ vg per mouse of AAV-hSPB plus BLES (pink line; $***P = 0.0004$) at P8 to P10 with dox removal at P21. Control mice were untreated and maintained on dox (black line; $**P = 0.0021$) or underwent sham surgeries with IT injections of 1× PBS plus BLES (blue line). **e** Mean weight measurements with SD of all surviving mice untreated and maintained on dox (black circles), treated with 1× PBS plus BLES with dox removed (blue squares), or treated with $10^{11}$ vg of AAV-hSPB plus BLES with dox removed (pink triangles) from **d**. (Survival curve $P$ values of **a** = 2-tailed Log-rank, Mantel-Cox test; Survival curve $P$ values of **d** = 2-tailed Log-rank, Mantel-Cox test with Bonferroni correction, significance is set at $P = 0.0167$, ns = not significant) (BLES = Bovine Lipid Extract Surfactant).

## Methods

**Institutional approval for animal and human studies**. All mouse experiments were performed in compliance with the guidelines set forth by the Canadian Council on Animal Care. All protocols and procedures involving animals were approved by the Animal Care and Veterinary Service Committee (ACVS) at the University of Ottawa, and the Animal Care Committee at the University of

Guelph. All human fetal tissues in this study were obtained after receiving institutional review board approval from The Ottawa Hospital (Research Protocol 20170603-01H), and after informed parental consent from donors.

**Recombinant AAV plasmid generation**. AAV genome plasmids contain the composite CASI promoter consisting of the human cytomegalovirus immediate

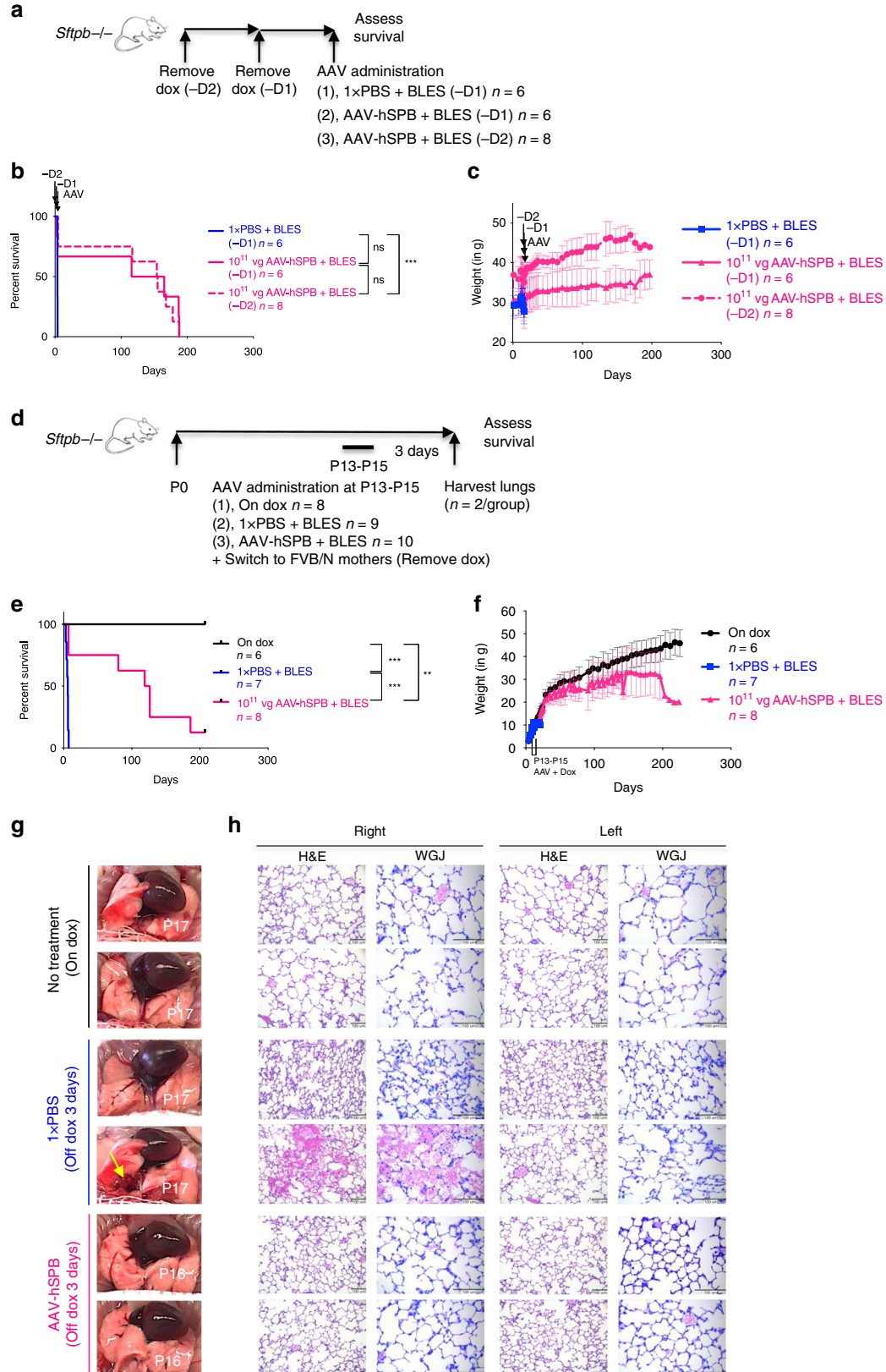

early gene enhancer region, the chicken beta actin promoter, and the human ubiquitin C promoter, as well as the woodchuck hepatitis virus posttranscriptional regulatory element (WPRE) and simian virus 40 polyadenylation sequence downstream of the transgene with flanking adeno-associated virus 2 (AAV2) inverted terminal repeats (ITRs)[48]. The murine *proSftpb* sequence (NM_147779.2) was synthesized (GenScript) after codon optimization for murine expression using the GenScript codon optimization tool, and contains a myc epitope tag at the

C-terminus. Human *proSFTPB* cDNA (NM_000542) was synthesized without codon optimization (GenScript) or an epitope tag.

**AAV6.2FF vector production**. All AAV6.2FF vectors were generated by plasmid transfection of adherent HEK293 cells using polyethylenimine at the University of Guelph[49], except for the AAV6.2FF-murine SP-B cDNA (AAV-mSPB) vector used

**Fig. 9 AAV-hSPB rescues respiratory distress in an inherited model of SP-B deficiency. a** Study design to assess survival following AAV-hSPB plus BLES gene therapy in inherited SP-B deficient adult mice. Control mice underwent sham surgeries with IT injections of 1× PBS plus BLES 1 day (-D1) after dox removal (blue line). Treatment mice had dox removed 2 days (-D2; dashed pink line) and 1 day (-D1; solid pink line) before IT administration of $10^{11}$ vg per mouse of AAV-hSPB plus BLES. **b** Kaplan-Meier survival curves from IT injections of a $10^{11}$ vg dose of AAV-hSPB plus BLES 1 (-D1) and 2 (-D2; ***$P$ = 0.0001) days after dox removal. **c** Mean weight measurements with SD of all surviving mice from **b**. **d** Study design to assess survival following IT delivery of $10^{11}$ vg per mouse AAV-hSPB plus BLES into P13 to P15 neonatal SP-B deficient mice. The pups were removed from dox immediately following AAV administration by placing them with wild type FVB/N foster mothers that had recently given birth to similar aged pups. Two mice were euthanized from each group 3 days after surgery for histological lung examination. **e** Kaplan-Meier survival curves following IT delivery of $10^{11}$ vg per mouse of AAV-hSPB plus BLES into P13 to P15 neonatal mice with dox removal immediately following surgery (***$P$ = 0.0004 compared to 1× PBS). Neonatal pups were untreated and maintained on dox (black line), underwent sham surgeries with IT injections of 1× PBS (blue line; ***$P$ = 0.0004 compared to on dox), or were IT administered $10^{11}$ vg of AAV-hSPB plus BLES (pink line; **$P$ = 0.0021 compared to on dox). The survival curve data do not include the mice removed for lung histology. **f** Mean weight measurements with SD of all surviving mice from **e**. **g** Gross lung images from all 3 groups ($n = 2$ per group) at harvest 3 days following (AAV administration plus dox removal. Yellow arrows indicate regions of lung injury. **h** Representative H&E and WGJ staining of paraffin embedded whole right and left lungs 3 days after AAV administration and dox removal from all 3 groups in **d** (H&E and WGJ scale bar, 100 μm). (All survival curve $P$ values in Fig. 9 = 2-tailed Log-rank, Mantel-Cox test with Bonferroni correction, significance is set at $P$ = 0.0167, ns = not significant).

in the intubation (Supplementary Fig. 8c) and high dose (Fig. 6b) survival experiments which was manufactured at the Senator Paul D Wellstone Muscular Dystrophy Cooperative Research Center vector core (Seattle, WA), and the AAV8-luciferase vector which was generated by the University of Pennsylvania vector core (Supplementary Fig. 7) (Philadelphia, PA). Vector genomes were quantified by TaqMan qPCR assay by targeting the inverted terminal repeat sequence of the AAV2 genome using the primers Forward: 5′–GGA ACC CCT AGT GAT GGA GTT–3′; Reverse: 5′ CGG CCT CAG TGA GCG A–3′; and Probe: 5′–FAM-CAC TCC CTC TCT GCG CGC TCG-BHQ – 3′[50].

**In vitro cell culture and transfections**. Human embryonic kidney 293 cells (HEK293; ATCC® CRL-1573) were maintained in high glucose DMEM (HyClone; SH30022.01) with 10% cosmic calf serum (HyClone; SH30087), 2 mM L-glutamine (HyClone; SH3003401), and 1% pen-strep (HyClone; SV30010). Murine lung epithelial 12 cells (MLE12; ATCC® CRL-2110) were maintained in DMEM/F12 1:1 media including HEPES (HyClone; SH30261) with the addition of 2% fetal bovine serum (FBS; HyClone; SH30088), insulin (0.005 mg per mL)-transferrin (0.01 mg per mL)-selenium 30 nM (Gibco; 41400045), 10 nM hydrocortisone (Sigma; H6909), 10 nM Beta-estradiol (Sigma; E2758), 2 mM L-glutamine (HyClone; SH3003401), and 1% pen-strep (HyClone; SV30010). At 75% confluency, HEK293 cells were transfected with linear polyethylenimine MW 25000, while lipofectamine was used to transfect MLE12 cells. Cell lysates were generated using a RIPA buffer (50 mM Tris pH 7.5, 150 mM NaCl, 1% Triton X-100, 0.1% SDS, 10 mM EDTA, 1% sodium deoxycholate) containing $Na_3VO_4$ (1 mmol per L), NaF (50 mM) and protease inhibitors (Sigma) 48 h post transfection, and prepared for immunoblotting using a 4 × SDS-PAGE reducing buffer. AAV-mSPB was incubated with HEK293 cells at a multiplicity of infection (MOI) of 20,000[24] and cells were harvested 72 h later and subsequently prepared for immunoblotting as described above.

**Immunoblotting**. Cell lysates were separated on a 10% tris-glycine polyacrylamide gel and transferred to a PDVF membrane at 100 V for 1 h. The membrane was blocked in 5% bovine serum albumin with PBS plus 1% Tween-20 (PBS-T) for 1 h while rotating, and washed 3 times with PBS-T. The membranes were incubated overnight at 4 °C with a c-Myc primary antibody at a 1 in 2000 dilution (Cell Signaling; D84C12), and washed 3 times with PBS-T before the addition of an anti-rabbit HRP polyclonal secondary antibody at a 1 in 4000 dilution (Invitrogen IgG (H + L); LS656120). Membranes were washed twice with PBS-T, once with PBS and imaged using HRP substrate (Luminata; WBLUF0100) in a BioRad Chemidoc. An uncropped version of the Western blot in Fig. 3b is supplied in the Source Data files.

**Animal model for AAV6.2FF targeting analysis**. The following procedure was carried out at the University of Guelph. Five 8-week old male Rosa26-Flox/LacZ (B6.129S4-Gt(ROSA)26Sor^{tm1Sor}/J; strain code 003474) mice purchased from Jackson Laboratory (Bar Harbor, ME) were housed at the University of Guelph in a specific pathogen-free isolation facility. The animals were maintained on a 12 h on and 12 h off light cycle, room temperature (RT) was 23 to 24 °C, and humidity ranged from 30% to 60%. Mice were housed in groups of 4 and food (Teklad Global 14% Protein Rodent Maintenance Diet) and water (tap) were provided ad libitum. Mice were acclimated to the environment for 7 days prior to study initiation.

Using a modified intranasal delivery method[37] or via tail vein injection, groups of 3 mice were administered $10^{11}$ vg of an AAV6.2FF vector expressing Cre recombinase containing a nuclear localization signal (nls) and fused to GFP (AAV6.2FF-CMV-emGFP-nlsCre) under the control of the CMV promoter. Control mice received 1 × PBS. At 3 weeks post-AAV administration, mice were humanely euthanized by isoflurane and all major organs harvested. Briefly, mice

were exsanguinated via heart puncture and perfused with 1 × PBS via the right ventricle of the heart to remove excess blood. Next, a 27-gauge needle was inserted into the cartilage rings of the trachea and 5 mL of fixative (0.5% glutaraldehyde, 2 mM $MgCl_2$, 0.02% NP40, 0.01% deoxycholate in 1 × PBS) was administered to fully inflate the lungs while still inside the rib cage to avoid over-inflation. The trachea and lungs were removed en bloc and fixed for 2 h at RT. The remaining organs including the heart, spleen, liver, kidneys, pancreas, and brain were fixed for 16 h at RT while shaking. After fixation, tissues were washed 6 times for 5 min in wash solution (2 mM $MgCl_2$ in 1 × PBS), before being stained in the dark with X-gal staining solution (5 mM potassium ferricyanide, 5 mM potassium ferrocyanide, 2 mM $MgCl_2$, 0.01% sodium deoxycholate, 0.02% Nonidet P-40 in 1 × PBS, pH 7.0 supplemented with 1 mg per mL X-gal [5-Bromo-4-chloro-3-indolyl-beta-D-galactopyranoside; 40 mg/mL stock in dimethylformamide] added immediately before use) for 2 h at 37 °C in the case of the lungs, and 8 h at 37 °C for all other tissues. Tissues were rinsed 6 times in 2 mM $MgCl_2$ in 1 × PBS until the solution no longer turned yellow, placed in a cryomold, embedded in optimal cutting temperature compound (OCT) and stored at −80 °C. Lung and tracheal tissue from mice receiving the vector intranasally was paraffin embedded, sectioned (4 μm) and counterstained with nuclear fast red. All other tissues were cryosectioned and re-stained with X-gal staining solution for 4 h at 37 °C in the dark and counterstained with nuclear fast red. Tissue sections were imaged using an Olympus BX45 light microscope.

**Animal model for lung analysis and survival studies**. The following procedures were carried out at The Ottawa Hospital Research Institute and The University of Ottawa. We utilized an inducible model of SP-B deficiency under the control of a doxycycline (dox) dependent promoter[16]. Mice were maintained on dox-supplemented feed (0.625 g per kg doxycycline hyclate; Teklad) unless otherwise noted. The animals were maintained on a 12 h on and 12 h off light cycle, RT was 21 °C, and humidity was 45%.

Mice were genotyped and maintained according to previously published protocols[5]. Briefly, 4 regions from the transgenic mouse model were genotyped. DNA extraction for PCR was obtained from clipped ear tissue using the AccuStart II Mouse Genotyping Kit (Quanta Biosciences) according to the manufacturer's instructions. For the rat CCSP promoter-rtTA genotyping–Forward: 5′-ACT GCC CAT TGC CCA AAC AC-3′; Reverse: 5′-AAA ATC TTG CCA GCT TTC CCC-3′; PCR Cycling Protocol: (1) 94 °C 5 min (2) 94 °C 30 s (3) 64 °C 30 s (4) 72 °C 40 s (5) Repeat 2-4 39 times (6) 72 °C 5 min (7) 4 °C. For the tetO7 minimal CMV promoter (murine *Sftpb* cDNA) genotyping–Forward: 5′-TGC TGC CAG GAG CCC TCT TG-3′; Reverse: 5′-AAG GCA CGG GGG AGG GGC AAA-3′; PCR Cycling Protocol: Same as the rat CCSP promoter-rtTA genotyping. For the phosphoglycerate kinase (Neomycin cassette) genotyping–Forward: 5′-TGA CCG CTT CCT CGT GCT TTA C-3′; Reverse: 5′-CCC CCC AGA ATA GAA TGA CAC CTA C-3′; PCR Cycling Protocol: (1) 94 °C 5 min (2) 94 °C 30 s (3) 64 °C 30 s (4) 72 °C 15 s (5) Repeat 2-4 39 times (6) 72 °C 5 min (7) 4 °C. For the murine SP-B exon 4 genotyping–Forward: 5′-CCA GGC TAA TCC TCC CTT CT-3′; Reverse: 5′-CCC ACT TAG GCA CAT GCA C-3′; PCR Cycling Protocol: (1) 94 °C 5 min (2) 94 °C 30 s (3) 62 °C 30 s (4) 72 °C 15 s (5) Repeat 2–4 39 times (6) 72 °C 5 min (7) 4 °C.

All mice were caged in either open or individually ventilated cage systems. Up to 4 animals were housed per cage and grouped with littermates and by sex. Mice were assigned a numerical ID. The treatment of mice (either control or AAV-mSPB and -hSPB) was assigned based on alternating numerical IDs. For example, in a cage of mice with the IDs L777, L778, L779, and L780: L777 and L779 would be administered AAV-mSPB while L778 and L780 would receive 1 × PBS (Fig. 6d). The order of surgeries were performed based on their numerical order: 1st: L777 (AAV-mSPB); 2nd: L778 (1 × PBS); 3rd: L779 (AAV-mSPB); 4th: L780 (1× PBS).

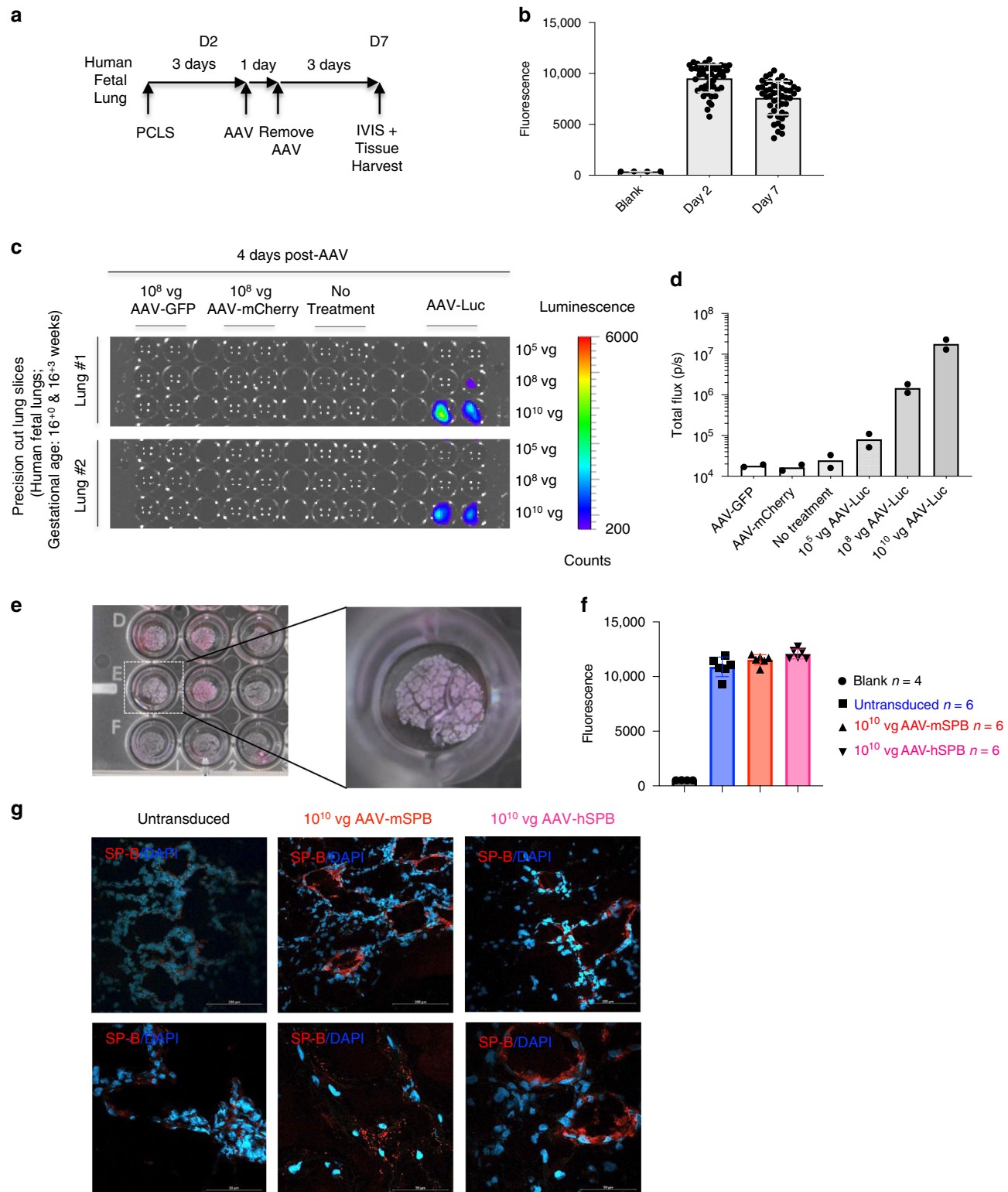

To study the effects of AAV-SPB in the animal model, mice were typically maintained on dox for 27 to 30 days (4 weeks) following AAV injection to allow sufficient time for transgene expression[26]. We also assessed the effects following dox removal 3 and 7 days after AAV administration, upon weaning (6 to 13 days after dox), and dox removal 1 or 2 days before AAV administration. Mice were regularly weighed before and after AAV delivery and dox removal. Treatment or negative control groups were taken off dox and placed on a regular chow diet (Teklad 2018, 18% protein rodent diet; Envigo) for 3 to 4 days before euthanizing and harvesting the lungs for structure and function analysis (Figs. 4a, 7b).

For neonatal survival studies, transgenic SP-B deficient mice at postnatal(P) ages P8 to P10 were either left on dox, treated with 1 × PBS + BLES, or treated with 10^11 vg per mouse AAV-hSPB + BLES by IT administration. At P21, the mice were weaned off dox (obtained through the milk of nursing mothers) onto a regular chow diet (except for the mice maintained on dox), and body weight and survival were recorded. The volume of fluid instilled into the lungs of P8 to P10 pups resulted in approximately 50% mortality due to the inability of their lungs to accommodate the injection volume (20 μL). Administration at P8 resulted in 7 out of 10 (70%) deaths, at P9 4 out of 11 (36%) deaths, and at P10 2 out of 5 (40%) deaths within 1 to 5 min of administration. Death occurred regardless of whether

**Fig. 10 AAV6.2FF transduces the human lung parenchyma. a** Study design to demonstrate the ability of AAV6.2FF to transduce human lung tissue samples generated from precision cut lung slices (PCLS). Metabolic viability (resazurin) assays were conducted on days 2 (D2) and 7 (D7) post tissue harvest. **b** Resazurin assay results as represented by mean fluorescence with SD demonstrating the metabolic activity of the human PCLS on days 2 and 7. **c** Bioluminescence detection using IVIS in PCLS from human fetuses ($n = 2$; $16^{+0}$ and $16^{+3}$ weeks gestational age) either untreated or transduced with $10^8$ vg per well of AAV-GFP, $10^8$ vg per well of AAV-mCherry, or $10^5$, $10^8$, or $10^{10}$ vg per well of AAV-Luc. Images are 4 days following vector exposure, and the scale is represented as counts. **d** Quantification of AAV-Luc wells are represented by the mean total flux (in photons per sec) with SD ($n = 2$ fetuses; with 12 replicates for the AAV-GFP, AAV-mCherry, and No Treatment groups; and 4 replicates for each of the AAV-Luc doses). **e** Images of human PCLS in a 24-well tissue culture plate 7 days after culture and 3 days after AAV-SPB transduction. An enlarged image of a PCLS in an individual well is indicated by the dashed white box. **f** Resazurin assay results as represented by mean fluorescence with SD demonstrating viability of the human PCLS 3 days after no treatment (blue), or transduction with $10^{10}$ vg per well of AAV-mSPB (red) or AAV-hSPB (pink). ($n = 4$ blank; and $n = 6$ untransduced, $n = 6$ AAV-mSPB, and $n = 6$ AAV-hSPB independent tissue slices from a single set of human fetal lungs). The source data for **f** has been provided as a Source Data file. **g** Representative confocal images of SP-B (red) and DAPI (blue) of frozen PCLS sections 3 days after transduction with $10^{10}$ vg per well of AAV-mSPB or AAV-hSPB (Top and bottom row image scale bars, 100 μm and 50 μm, respectively).

1 × PBS or AAV-hSPB was instilled. To reduce post-surgery death, we repeated this experiment using older neonatal pups (P13 to P15) which significantly reduced the mortality (1 out of 20 deaths or 5%). In this latter experiment, pups were removed off dox immediately following AAV administration by placing them with nursing FVB/N foster mothers (on a regular chow diet) who had recently given birth to similar age-matched litters. This required the setting up of breeding pairs to deliver litters at the same time from both transgenic SP-B deficient mice on dox (6 pairs), and FVB/N mice on a regular chow diet (6 pairs). The inducible transgenic SP-B mouse model was generated by injecting murine *Sftpb* cDNA under the control of a (teto)$_7$ promoter into fertilized FVB/N oocytes[16].

For survival studies mice were monitored at least 2 times per day following dox removal. When mice began to display initial signs of respiratory distress and weight loss, monitoring took place every 2 to 3 h. A scoring and endpoint chart for signs of respiratory distress requiring euthanasia was provided by the ACVS at The University of Ottawa (Supplementary Data File 2). Once respiratory distress was identified, animals were euthanized within 1 h of observing signs and symptoms of M2, and were immediately euthanized upon displaying signs and symptoms of M3. The decision for identifying the respiratory signs and symptoms for euthanasia were made by the staff at ACVS or by members of our group blinded to the identity and treatment groups of the animals. Mice were euthanized by 100% carbon dioxide ($CO_2$) inhalation.

Although it was reported that the inducible SP-B deficient mice suffer lethal respiratory distress at 7.5 days ± 3.5 days (4 to 11 days) off dox[16], the majority of untreated mice suffered lethal respiratory distress before 4 days. We also observed a few cases ($n = 3$) where untreated mice survived beyond their expected endpoints (Supplementary Fig. 6). To ensure that AAV-SPB improvements in survival or lung structure and function were not due to stochastic incidences of extended survival, negative control groups were almost always used in our studies.

**Intratracheal (IT) injections.** One hour before IT surgery, animals were injected subcutaneously (SC) with 0.1 mg per kg buprenorphine (Champion Alstoe). Just before surgeries, animals were individually anesthetized with isofluorane (Fresenius Kabi) and injected SC with 1 mL of sterile 0.9% sodium chloride saline solution (Baxter Corp). The fur covering the tracheal region was then shaved to the skin. A tracheotomy was performed under constant isofluorane anesthesia, with mice receiving a single IT injection of AAV vector. Different titers of AAV-mSPB ($10^{10}$ vector genomes (vg); $10^{11}$ vg; or $5 × 10^{11}$ vg) or AAV-hSPB ($5 × 10^{10}$ vg; $10^{11}$ vg) were diluted to a total volume of 42 to 83 μL with 1 × PBS or pulmonary surfactant (Bovine Lipid Extract Surfactant; BLES Biochemicals Inc). Injections were carried out using a 3/10 mL insulin syringe 29 gauge × ½" (Covidien). Openings were sutured and topical 2% transdermal bupivacaine HCl as monohydrate (Chiron) was applied on the surgical site post-surgery, and 4 to 6 h after surgery. Mice were allowed to recover in a 37 °C incubator for 1 h post-surgery.

**Intubation (Int) injections.** Intubations were performed based on a combination of previously described methodologies[37,51]. Briefly, mice were injected SC in the posterior dorsal region with 1 mL of saline to prevent dehydration. Mice were then anesthetized intraperitoneally (IP) with 100 mg Ketalean (Ketamine; Bimeda-MTC Animal Health Inc) per 10 mg Rompun (Xylazine; Bayer Inc) per kg of body weight. Mice older than 6 weeks of age and weighing 20 to 35 g were intubated with a 1 inch long, 22 gauge IV catheter (BD Insylte). To view the tracheal opening, approximately 60 cm of 0.5 mm optical cable (Edmund Optics) was used with one end inserted through the 22-gauge IV catheter and the other end inserted through a rubber stopper connected to a light source. The anesthetized animal was placed on a vertical support (Harvard Apparatus) and suspended by its upper incisors using surgical thread. The ventral side of the mouse was positioned facing away from the operator and the tongue was gently pulled out using forceps. The fiber optic cable plus IV catheter was inserted with the light source used to visualize the vocal cords. The cable plus catheter was advanced past the vocal cords into the trachea, the fiber optic cable was withdrawn, and a 1 mL syringe containing 50 to 60 μL of AAV

vector was administered through the catheter. This was followed by three 100 μL injections of air with the 1 mL syringe, in an attempt to improve AAV distribution throughout the distal lung. The mouse was left in this standing or upright position for 1 to 2 min to allow time for vector dispersion throughout the lungs. Eye lube (CLC Medica) was continuously applied to protect against corneal drying and injury. Mice were allowed to recover for 1 to 2 h in a 37 °C incubator post-surgery.

**In vivo imaging system and diffuse light imaging tomography.** Mice were administered with a single injection of $10^{11}$ vg of AAV6.2FF-Luciferase (AAV-Luc) or AAV8-Luc. Beginning 1-week post injection, IVIS or DLIT images were acquired. D-luciferin, Sodium Salt (BioVision) was sterilely prepared 24 h before injection at a concentration of 15 mg per mL in 1 × PBS. All mice were SC injected with 150 mg per kg of D-Luciferin 15 to 20 min before IVIS imaging. Up to 4 mice were imaged by IVIS simultaneously, while DLIT was performed on a single mouse. For DLIT imaging, surface topography was generated, followed by 3D reconstruction of bioluminescence as per the manufacturer's instructions (PerkinElmer). All DLIT images were performed within 30 to 45 min of D-Luciferin injection.

All quantifications of regions of interest (ROI) were generated using the Living Image 3.2 software according to the manufacturer's instructions (Caliper LifeSciences). Briefly, each mouse was identified through the generation of subject ROIs automatically by the Living Image software. Subsequently, bioluminescence signal intensity was manually identified in each mouse by an operator. ROIs larger than the visual boundaries of the bioluminescence signal were defined to ensure that the entire area of signal diffusion was included in each measurement. For each animal, the total flux (photons per second; p per s), average radiance (photons per second per square centimeter per solid angle of 1 steradian; p per s per $cm^2$ per sr), the standard deviation (SD) of radiance, as well as the minimum and maximum radiance were obtained from the nasal and tracheal, thorax (lung), and abdominal regions. All images are presented as count measurements which are uncalibrated measurements of photons in a pixel. Counts allow normalization between images acquired at different times. The total flux (p per s) or radiance (p per s per $cm^2$ per sr) was used to quantitate bioluminescence signal from IVIS figures in this study, which are calibrated measurements of photon emissions.

**Generation of precision cut lung slices (PCLS).** PCLS were processed based on previous descriptions[21]. Briefly, mice were euthanized with an IP injection of Euthanyl (Pentobarbital Sodium Injection; Bimeda-MTC Animal Health Inc), and the lungs were instilled with a low gelling point agarose solution (0.7% weight per volume [w per v] in 1 × PBS) under a constant pressure of 20 cmH$_2$O. After cooling the mice in an ice bath, the left lung was isolated and embedded in a 5% low gelling point agarose solution.

Human fetal samples were obtained after elective abortions. The lungs were isolated by a pathologist and transported to the laboratory in ice cold DMEM GlutaMAX (Gibco; 10569-D10). Samples were rewarmed in 37 °C DMEM GlutaMAX for 30 min before low gelling point agarose solution (0.7% to 1.5% weight per volume) was slowly instilled into the lungs via the trachea or the main bronchus. Instillation was terminated when the lungs were inflated and the agarose solution overflowed from the airway. After cooling in an ice bath, the lungs were sampled and embedded in a 5% low gelling point agarose solution.

The tissue slicer (Kruldieck Tissue Slicer, TSE Systems) was set to a slicing thickness of 300 μm at a speed of 30 slices per minute, and PCLS were collected in lung slice wash medium (LSWM: DMEM GlutaMAX (Gibco; 10569-D10), Anti-Anti (Gibco; 15240-062), gentamycin (ThermoFisher; 15710-064), 10 μM 8-bromo-cAMP (Sigma; B5386), 100 μM IBMX (Sigma; D4902), 100 nM Dexamethasone (Sigma; D4902)). The PCLS were maintained in a 48-well tissue culture plate (1 slice per well) in lung slice maintenance media (LSMM: LSWM with 10% FBS (Sigma; F1051), 2 mM L-glutamine (Gibco; 25030-081), 10 ng/mL recombinant KGF (FGF-7) (ThermoFisher; PHG0094)). The PCLS were acclimated for 3 days with fresh LSMM added daily. The viability of the PCLS were

assessed by performing resazurin (Sigma; 199303) fluorescence assays which was used to measure metabolic activity on the day of tissue collection (day 0), before or on the day of AAV transduction (day 2 or 3), and the day of IVIS imaging (day 7). On day 3, the adult mice or human fetal PCLS were transduced with $10^8$ vg per well of AAV6.2FF-GFP, -mCherry, -Luc, -mSPB, or -hSPB. In the human fetal PCLS, 3 different AAV-Luc titers were used ($10^5$, $10^8$, and $10^{10}$ vg per well), and $10^{10}$ vg per well of either AAV-mSPB or AAV-hSPB was used. Untransduced PCLS were used as a negative control. The viral vectors were applied to each well in a total volume of 150 to 300 µL of neat DMEM GlutaMAX (depending on their size) for 24 h with constant rocking. Seven days after initial collection of PCLS and 4 days after vector exposure, the PCLS underwent IVIS imaging and were collected in 4% PFA, or embedded in VWR Premium Frozen Section Compound (VWR; 95057-838) and frozen on dry ice.

**Flow cytometry to determine cell-types transduced by AAV.** Eight days after IT administration of $10^{11}$ vg of either AAV-Luc or AAV-GFP, mice were IP injected with 1 mUnits per g of heparin, euthanized by Euthanyl (Pentobarbital Sodium Injection; Bimeda-MTC Animal Health Inc), and had the thorax region surgically opened. The lung vasculature was perfused with 5 to 10 mL of cold $1 \times$ PBS (no $Ca^{2+}$ and $Mg^{2+}$) plus undiluted heparin (25 µL per 1 mL) through the right atrium to remove red blood cells (RBC). Following perfusion, an enzyme mix containing 30 U neutral protease (Worthington; LS02104), 2500 U Collagenase I (Worthington; LS004196), 10 µg DNase I (Sigma; D5025-150KU) in DPBS with $Mg^{2+}$ and $Ca^{2+}$ was instilled into the lungs. The lungs were removed and placed in 5 mL of the enzyme mix for 1 h at 37 °C. Following removal of all traces of the trachea, bronchi, and heart tissue, the lung lobes were dissociated into small pieces and passed through a 40 µm cell strainer. The filtered lung suspensions were collected and washed in FACS buffer ($1 \times$ PBS, 5% FBS (Sigma; F1051), 1 mM EDTA). All centrifugation steps were carried out at 500 rcf for 5 min. Cold RBC lysis buffer was added to the cells and washed 3 times before the total cell number was counted for each sample. A total of $5 \times 10^5$ cells in 200 µL were placed into individual wells of a 96-well plate for staining. A pooled lung sample from the remaining AAV-Luc and AAV-GFP suspensions was used as a control at a concentration of $10^5$ cells in 200 µL. All samples were blocked with Heavy chain (Fc) block at RT in the dark. The samples were resuspended in 100 µL of FACS buffer with the following antibody dilutions: 1 in 100 CD31-BV421 (Biolegend; Cat#102424); 1 in 100 CD45-AF647 (Biolegend; Cat#103124); and 1 in 100 EpCam-PE-CY7 (Biolegend; Cat#118216), and incubated for 30 min. The cells were washed 3 times with FACS buffer, and then resuspended in FACS buffer with 1 µL of 7-AAD (Biolegend; Cat#420403) for 5 to 10 min before flow cytometry. No stain, single stain, and fluorescence minus one (FMO) control stains were all carried out. The samples were analyzed on the BeckmanCoulter MoFlo XDP flow cytometry machine (BeckmanCoulter), sample compensation was performed with the BD FACSDiva software version 8.0, and data was analyzed using FlowJo version 10 (Becton, Dickinson and Company).

**Lung function analysis.** Lung function including a measure of pulmonary surfactant function was calculated following the generation of individual pressure-volume (PV) curves for each mouse[52,53]. Mice were euthanized with an IP injection of Euthanyl (Pentobarbital Sodium Injection; Bimeda-MTC Animal Health Inc). Within 10 to 15 min of Euthanyl injection, pressure-volume curves were obtained using a small animal mechanical ventilator (flexiVent, Scireq). Briefly, an 18-gauge cannula attached to the flexiVent was secured to the trachea of euthanized animals in a supine position. The lungs were inflated with regular increasing intervals of pressure to a maximum of 30 cmH$_2$O. Lungs were subsequently deflated with regular decreasing intervals of pressure to obtain pressure-volume curves. All data were obtained using the flexiWare version 7.0 (Scireq) software. Pressure-volume curves were normalized to the body weight of each animal.

%V$_{10}$, total lung capacity (TLC), residual volume (RV), and lung compliance were extracted from the pressure-volume curves[31]. Briefly, %V$_{10}$ quantifies the shape of the deflation limb and decreases with surfactant inhibition and is calculated using Eq. (1). The TLC is the volume at the defined maximal pressure of 30 cmH$_2$O. The RV is the residual air trapped in alveoli at maximal expiration at the end of the deflation curve. The compliance is the slope at any linear region of the deflation limb and was calculated by the ratio in the change in volume between 10 cmH$_2$O and 7 cmH$_2$O ($\Delta V$) over the change in pressure ($\Delta P$).

**Quantitative PCR expression analysis of lung tissue.** RNA was isolated from snap frozen right lung tissue using QIAamp® Viral RNA Mini Kit (Qiagen). A total of 1 µg of DNase treated RNA was reverse transcribed into complementary DNA (cDNA) using reverse transcriptase (Invitrogen). Real time qPCR (SYBR Green) was performed using the following primers for proSP-B-Myc (Murine proSP-B Forward: 5′-CCA AGA TCT CAG GAC GCC-3′; Myc Reverse: 5′-TCA CAG ATC CTC TTC TGA GAT G-3′) and for Gapdh (Gapdh Forward: 5′-GTT GTC TCC TGC GAC TTC A-3′; Gapdh Reverse 5′-GGT GGT CCA GGG TTT CTT A-3′). All qPCR reactions were carried out on a LightCycler 480 II (Roche Diagnostics) with Ct values automatically generated using the StepOnePlus software 2.3.

**Lung histology.** Macroscopic lung images were obtained with an iPhone 6S camera (Apple), or a Canon PowerShot SX620 HS point-and-shoot camera. For

microscopic lung images, lungs were perfused with 4% paraformaldehyde (4% PFA; Sigma-Aldrich) and fixed for 2 days. On day 3, the 4% PFA was removed and replaced with 70% ethanol. The left lungs were embedded in paraffin and cut coronally to obtain 4 µm longitudinal sections of the lung. Serial sections were stained with hematoxylin and eosin (H&E) or Wright-Giemsa Jenner (WGJ) by the University of Ottawa histology core (Ottawa, ON). Scanned images were obtained at ×20 with the Aperio CS2 digital brightfield scanner (Leica) using Aperio's ImageScope version 12.3 software, and at ×20 and ×40 with the Leica DM4000 upright brightfield microscope.

**Lung immunofluorescence (IF).** All IF images were obtained from lung sections perfused with a 1 to 1 ratio of $1 \times$ PBS and VWR Premium Frozen Section Compound (VWR; 95057-838), frozen on dry ice and stored at −80 °C for at least 24 h. All sections were cut into 6 to 8 µm sections using the Leica CM1860 cryostat. Sections were air-dried for 3 h at RT and stored at −20 °C for at least 24 h. Prior to fixing and staining, frozen sections were thawed at 37 °C for 2 to 3 h. Briefly, sections were fixed in −20 °C acetone for 15 min and washed with 0.1% Tween-20 in $1 \times$ PBS 2 times (5 min per wash) and $1 \times$ PBS once (5 min). Antigen retrieval in 10 mM sodium citrate pH 6.0 solution with 0.05% Tween-20 heated to boiling in a microwave was carried out 3 times (10 min per incubation). The sections were permeabilized in 0.1% Triton X-100 in $1 \times$ PBS for 10 min at RT and blocked in 10% FBS in $1 \times$ PBS for 1 h at RT. Sections were stained with a dilution of 1 in 250 proSP-C (rabbit anti-proSP-C; Millipore Sigma; AB3786) or 1 in 500 SP-B (rabbit anti-SP-B; Seven Hills Bioreagents; WRAB-48604) primary antibodies for 2 to 3 days at 4 °C in a light-resistant slide box. Human fetal PCLS were stained with a dilution of 1 in 50 firefly luciferase (goat anti-luciferase; Novus; NB100-1677), 1 in 200 mCherry (mouse anti-mCherry; Novus; NBP-1-96752), 1 in 50 EpCAM (rabbit anti-EpCAM; abcam; ab71916), 1 in 250 proSP-C, or 1 in 500 SP-B primary antibodies. Secondary antibody staining was with a dilution of 1 in 500 of goat α-rabbit AF-568 (Life technologies; A11011), 1 in 500 of donkey α-rabbit AF-488 (Invitrogen; A21206), 1 in 500 of goat α-mouse AF-568 (Invitrogen; A21124), or 1 in 500 of donkey α-goat AF-568 (Invitrogen; A11057) for 1 h at RT. Final washing steps were with $1 \times$ PBS 3 times (5 min per wash) and coverslips were mounted on sections with Fluoroshield with DAPI (Sigma; F6057). ×20 images were obtained with an epi-fluorescence microscope (Zeiss Axio Imager.M2) using AxioVision 40 × 64 version 4.9.1.0. Confocal images of human PCLS transduced with $10^{10}$ vg per well of AAV-mSPB or AAV-hSPB were obtained by the Zeiss LSM800 AvioObserver Z1 microscope using Zen 2 (Blue edition).

**Transmission electron microscopy (TEM) of AT2 cells.** Lung tissue for imaging by TEM was processed based on previous descriptions[54]. Briefly, the right bronchus was tied off and a fixative solution containing 1.5% glutaraldehyde (Sigma-Aldrich) and 1.5% PFA in 0.15 M HEPES buffer (Sigma-Aldrich; 83264) was instilled into the left lung. The trachea was tightly tied to ensure the fixative remained in the lungs, before the left lung was excised and placed into 15 mL of the fixative solution and stored at 4 °C. Lung samples were subsequently processed as rapidly as possible. The lungs were incubated with 1% osmium tetroxide, stained en bloc with half-aqueous uranyl acetate, dehydrated in an ascending acetone series, before embedding in epoxy resin. From the embedded samples, ultrathin sections (60 to 80 nm thickness) were cut, mounted on formvar-coated copper support grids, and post-stained with lead citrate and uranyl acetate. The sections were analyzed using a Morgani TEM (Field Electron and Ion Company, FEI).

**Pulmonary surfactant as a vehicle for AAV vector delivery.** Clinical grade exogenous pulmonary surfactant Bovine Lipid Extract Surfactant (BLES; BLES Biochemicals Inc) enriched for hydrophobic phospholipids (27 mg phospholipids/mL of suspension) and the SP-B and SP-C surfactant proteins was used as a vehicle for AAV vectors. Clinically, BLES is used to treat neonatal respiratory distress syndrome at a recommended concentration of 135 mg phospholipids per kg (BLES Biochemicals). BLES was administered by IT or intubation injections at various concentrations including: 1.34 mg of total phospholipids per mouse (Supplementary Fig. 7), 0.54 mg of total phospholipids per mouse (Supplementary Fig. 8a), 1.21 mg phospholipids per mouse (Supplementary Fig. 8c), 0.82 mg of total phospholipids per mouse (Fig. 8a), 0.92 mg of total phospholipids per mouse (Figs. 8a, 9b), or 0.11 mg of total phospholipids per mouse (Figs. 8d, 9e). Differences in BLES concentration between the experiments were due to differences in viral titers from individual AAV preparations, combined with attempts to maintain instillation volumes at approximately 50 µL per mouse. In the intubation plus BLES group (Supplementary Fig. 7), 1 mouse died within 10 min of administration of the vector plus BLES, presumably due to difficulties in respiration following fluid instillation into the lungs.

**Inflammatory cytokine measurements from mouse serum.** Serum was collected from the lateral saphenous vein of SP-B deficient mice in a survival study 7 days before (−7D), and 1 (+1D), 7 (+7D), and 29 days (+29D) after AAV or LPS administration in each of the following three groups: $1 \times$ PBS ($n = 7$) by IT administration maintained on dox; $10^{11}$ vg AAV-mSPB ($n = 8$) by IT administration with dox removal 7 days after AAV treatment; and 3 mg per kg of lipo-polysaccahrides (LPS) from *Escherichia coli* O111:B4 (Sigma; L2630) ($n = 8$) by a

single bolus IP injection maintained on dox. The 3 mg per kg dosage of LPS is considered sublethal in adult mice[40], however 5 out of 8 mice died post-LPS administration. Thirteen humoral factors were analyzed by multiplex technology using the LEGENDplex (BioLegend) mouse inflammation panel according to the manufacturer's instructions. This assay is a bead-based immunoassay operating in a similar principle to the sandwich immunoassay. Briefly, the samples were analyzed in a V-bottom plate on a BD FACSCanto II flow cytometer using a high throughput autosampler, and data was collected using BDFACSDiva version 8.0. All statistics were performed using the Mixed Effects Model with the Geisser-Greenhouse correction followed by a post hoc Tukey's multiple comparison's test. Note that data was analyzed by fitting the Mixed Model rather than by repeated measures ANOVA as there were missing values due to the death of animals (5 LPS and 1 AAV-mSPB mice) during the course of the study.

**Statistical analyses and reproducibility**. Graphpad Prism 8 software was used to perform all statistical analyses. Graphically, means are always presented with standard deviation (SD) error bars. For 2 groups, statistical analysis was by the 2-tailed Student's $t$-test. For 3 groups, statistical analysis was by ordinary one-way ANOVA with Tukey's or Dunnet's multiple comparisons post hoc test. Kaplan Meier survival curves were analyzed with the Log-rank, Mantel-Cox test. When more than 2 variables were compared in survival studies, Bonferroni correction was applied. Cytokine measurement data were analyzed by fitting the Mixed Effects Model with Tukey's multiple comparison post hoc test. A $P$ value less than 0.05 was considered significant, unless Bonferroni correction was applied. All instances of Bonferroni correction and significant $P$ values are indicated in the manuscript text or figure legends.

For representative experiments such as micrographs, the following list provides the number of times an experiment was independently repeated for a given figure: Fig. 2b ($n = 2$); Fig. 3b ($n = 3$); Fig. 4e ($n = 10$ for all groups except AAV-Luc, $n = 7$); Fig. 5b ($n = 2$ per group); Fig. 5c ($n = 2$ per group); Fig. 7f ($n = 2$ per group); Fig. 7g ($n = 2$ per group); Fig. 9h ($n = 2$ per group); and Fig. 10g ($n = 6$ PCLS slices per group). For the supplementary figures: Supplementary Fig. 1c ($n = 3$); Supplementary Fig. 2c (different field of view from image in Fig. 2b); Supplementary Fig. 2d ($n = 1$); Supplementary Fig. 3i ($n = 2$); Supplementary Fig. 4 ($n = 1$); Supplementary Fig. 5d ($n = 10$ for all groups except AAV-Luc, $n = 7$); Supplementary Fig. 5f–h ($n = 3$ for all groups); Supplementary Fig. 5j (from the same experiment as Fig. 5c, $n = 2$ per group); and Supplementary Fig. 6e, f ($n = 1$).

**Reporting summary**. Further information on research design is available in the Nature Research Reporting Summary linked to this article.

## Data availability

The authors declare that the data supporting the findings of this study are available within the paper, the associated Supplementary Figures and Tables, and in the Source Data files. Source data are provided with this paper. All other information is freely available from the corresponding authors upon request.

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

## Acknowledgements

This work was supported in part by funding from a CIHR Foundation Scheme Grant to B.T. (FDN-143316), a CHIR Project grant to S.K.W. and B.T. (420518), grants from The Lung Association–Ontario (34998) and The Mason Research Fund (44610) to S.K.W., NIH grant support to J.A.W. (U01HL134745 and U01HL148856), a European Respiratory Society Fellowship to L.R. (LTRF 2018), and Deutsche Forschungsgemeinschaft funding to I.M. The authors thank all the ACVS veterinary and technical staff at the University of Ottawa, and in particular Dr Mark Liepmann for the development of the respiratory distress scoring chart; Dr Greg Cron, for training of IVIS imaging; all the staff at the University of Ottawa Histology Core, especially Zaida Ticas and Sharlene Faulkes; Jeff McClintock at the CHEO Electron Microscope lab; and EORLA staff for the human fetal lung sample isolation, especially Janet Stinson and Korrine Hutt-Acres.

## Author contributions

M.H.K., S.K.W., and B.T. designed all the experiments and drafted the manuscript. M.H.K., L.P.v.L., L.X., J.M.D., A.V., L.R., C.M., M.H., I.M., Y.P., J.P.v.V., S.P.T., C.M., and C.C-D. performed all the experiments. L.P.v.L., J.M.D., Y.P., and S.K.W. designed, generated, and purified the AAV vectors. M.H.K., L.X., A.V., L.R., M.H., I.M., C.M., and C.C-D. performed all the in vivo experiments. L.X. maintained and genotyped all transgenic mice used in this study. A.V. performed all the IT surgeries. L.X. performed all the intubation surgeries. M.H.K. and L.X. designed and performed all the IVIS and DLIT studies. L.R. designed and generated the PCLS studies and results. C.M. processed and generated all the TEM images. M.H., and I.M. designed and generated the flow cytometry analyses. Y.P. performed the qPCR analysis. J.P.v.V. performed the cytokine measurements. S.P.T. performed the Cre recombinase studies. M.H.K., L.X., and C.C-D. performed the flexiVent analyses and harvested the lungs for histology. M.H.K. and C.M. generated all the immunofluorescence images. M.H.K. and L.X. monitored the mice for respiratory distress in the survival studies, and regularly recorded their body weights. L.P.v.L., L.X., J.M.D., A.V., L.R., C.M., M.H., I.M., J.A.W., L.M.N., S.K.W., and B.T. provided critical comments and suggestions during the drafting of the manuscript. J.M.D and A.V. contributed equally to this work.

## Competing interests

S.K.W., L.P.v.L., B.T., and M.H.K. have a patent pending entitled: 'Adeno-Associated Virus Particle With Mutated Capsid And Methods Of Use Therof.' The remaining authors declare no competing interests.
