## [Peer Review File · Nature Communications]

Reviewers' Comments:

Reviewer #1:

Remarks to the Author:

The results using AAV6.2FF for gene transfer and durable expression of SPB are profound and could be of clinical utility for patients with SPB deficiency as a bridge to transplant. The authors have shown that the AAV6.2FF variant is able to transduce both AT2 cells and ciliated epithelial cells in the lung with reasonable efficiency. When either mouse or human SPB is cloned into the vector, both can rescue lung injury, SPB expression, and prolong survival in a dose-dependent manner in a compound transgenic mouse model of SPB deficiency. Survival can be increased up to at least 180 days in these mice that otherwise die at 7 days post-dox withdrawal. While several other reports have shown that gene transfer and expression of either modified mRNA or plasmids expressing SPB can rescue the SPB phenotype in these mice, the maximal survival was much less using these other approaches, suggesting that the AAV6.2FF vector is more efficient and perhaps closer to clinical utility for this indication. In looking toward clinical use, the authors have also shown that this virus can transduce human lung *ex vivo*. Studies are sufficiently powered and described.

Several issues are somewhat concerning with the results in this manuscript. First is that both ciliated epithelial cells and ATII cells appear to be targeted by AAV6.2FF. A more detailed analysis of what these ciliated cells are would be valuable. Further, only FACS data is provided to support the "cell-specificity" of the GFP virus. Immunofluorescent staining of thin sections from mice given AAV6.2FF-GFP showing distribution of delivery and GFP transgene expression would be of value. While they stain for SPB expression in SBP compound transgenic mice in following doxycycline withdrawal, staining is difficult to visualize and it appears that the number of SPB positive cells is much less than 21.5% of ATII cells as stated for AAV6.2FF-GFP FACS analysis. A second concern also focuses on the non-ATII cells that are targeted by AAV6.2FF. If these cells are indeed targeted by the virus, what is the effect of SPB overexpression in cells that do not normally express SPB? How would this affect any translation to the clinical setting? A third concern is one of cell-specificity of this virus. While the authors show no endothelial cell transduction in the lung following intratracheal delivery, this is not surprising given that endothelial cells are not available to the virus by this delivery route. It could be valuable to explore the "cell-specificity" of this virus when delivered by various routes (IV, IM, etc) or in cultured cells other than pulmonary epithelium.

Reviewer #2:

Remarks to the Author:

Kang et al. here describe the use of a lung tropic AAV vector to deliver a murine Surfactant Protein B (SP-B) construct to improve survival and surfactant function in a mouse model of SP-B deficiency. Using a CMV gene enhancer region, a chicken β -actin promoter and either myc-tagged SP-B construct or a luciferase reporter, the authors show that the AAV vector mediates long-term gene expression that appears to be specific to EpCAM positive lung epithelial cells with an efficiency of approximately 20%. The mouse model of SP-B deficiency is one where a doxycycline-driven SP-B construct is expressed in a SP-B deficient mouse. Thus, removal of doxycycline results in SP-B deficiency and fairly rapid death in the absence of treatment. Administration of AAV-SP-B a month before removal of doxycycline was associated with robust expression of SP-B in Ep-CAM+ cells in the absence of doxycycline and loss of SP-B transgene expression, with evidence of lung protection from acute injury and maintenance of lamellar bodies in alveolar type 2 (AT2) cells. Additionally, PV curves, deflation stability and compliance were improved in mouse lungs treated long-term with AAV-SP-B. Studies to determine dose-dependence and the time course for transgene expression showed that increasing dose of AAV-SP-B was associated with increased length of survival that was unaffected by administration of the vector with exogenous surfactant. Long-term AAV-SP-B expression prevented lung injury and surfactant dysfunction from SP-B

deficiency. Finally, the authors demonstrate that this vector can transduce precision cut human lung slices from human fetal lungs. The manuscript is well written, extensive data with appropriate controls are presented, and the data are internally consistent. The following issues arise from review of this manuscript:

1. While the model used to induce SP-B deficiency is established and results in profound SP-B deficiency, it does not mimic the human condition appropriately. Human infants with congenital SP-B deficiency are never exposed to SP-B and develop respiratory distress immediately after birth. Identification of these infants for gene therapy is difficult as the thought that this could be a genetic defect of surfactant is not high in the differential diagnoses in neonates with acute respiratory failure at birth. Further, the current studies are unable to address the effect of SP-B exposure in a previously naïve state with respect to the induction of anti-SP-B antibodies by the host. This could alter the efficiency of transgene expression and, more importantly, hamper future lung transplantation to correct the abnormality. The experiments conducted are also in adult and not in neonatal mice who have not yet completed full lung development, in particular alveolarization. The dose response to AAV-Luc and AAV-SP-B needs to be evaluated in neonatal mice with particular attention paid to the effects on inflammation and alveolarization. Further, in order to be translatable to the human condition, it is possible that in utero therapy would need to be achieved. Perhaps the use of mice heterozygous for the SP-B gene should be used for in utero or early neonatal treatment studies to validate this approach. Thus, the authors' contention that the studies described here will result in rapid translation to the human appears to be premature. Indeed, the model is described as one of inherited SP-B deficiency. This is not accurate as this model is a loss of SP-B and more akin to acquired SP-B deficiency. This should be clarified in the manuscript, especially the title.

2. The effects on inflammation have not been addressed in any of the experimental conditions examined. This should be included in the manuscript for the long-term and day 3 and 7 experiments.

3. The authors show EpiCAM+ cells being preferentially transfected with AAV-SP-B. However, the efficiency of transfection is only demonstrated for the AAV-Luc construct. The authors should demonstrate the efficiency of AAV-SP-B transfection by determining the number of SP-B positive cells as a proportion of total Epi-CAM+ cells.

4. The authors contend that the administration and effects of AAV-SP-B were enhanced by using exogenous surfactant delivery (BLES). However, Fig.3d and Suppl Fig 6 do not support this conclusion.

5. Interestingly, the effects of AAV-SP-B on survival are better in a model in which doxycycline is removed three or seven days after AAV-SP-B administration. This could be defined as the time to 50% mortality which appears to be 125 days in the model where AAV-SP-B is administered a month before doxycycline withdrawal and 150 days in the model where the transgene is administered 3 or 7 days before doxycycline withdrawal. Could this indicate a long-term adverse effect of the AAV vector?

6. Human lung precision cut slices were treated with AAV-Luc or AAV-mCherry constructs. The effects of the AAV-SP-B construct should also be shown to demonstrate the cell types that are transduced by such treatment in human tissues. Can an alveolar type 2 phenotype be driven by expression of SP-B in fetal human lung under these conditions?

7. In Suppl Fig 5c, why was 5×10^{11} vg/mouse used in these experiments as this was the optimal dose in Fig 3?

8. The quality of the immunofluorescence and gross anatomy figures is poor. It is almost impossible to see the immunofluorescence and the gross pathology of the lungs are dark in all

figures. In some images, the DAPI staining is also difficult to see (e.g. Suppl Fig 3). All these figures need to be improved substantially.

9. The experimental methods are sufficiently detailed and appropriately catalogued.

10. The statistical methods are appropriate.

Reviewer #1:

The results using AAV6.2FF for gene transfer and durable expression of SPB are profound and could be of clinical utility for patients with SPB deficiency as a bridge to transplant. The authors have shown that a the AAV6.2FF variant is able to transduce both AT2 cells and ciliated epithelial cells in the lung with reasonable efficiency. When either mouse or human SPB is cloned into the vector, both can rescue lung injury, SPB expression, and prolong survival in a dose-dependent manner in a compound transgenic mouse model of SPB deficiency. Survival can be increased up to at least 180 days in these mice that otherwise die at 7 days post-dox withdrawal. While several other reports have shown that gene transfer and expression of either modified mRNA or plasmids expressing SPB can rescue the SPB phenotype in these mice, the maximal survival was much less using these other approaches, suggesting that the AAV6.2FF vector is more efficient and perhaps closer to clinical utility for this indication. In looking toward clinical use, the authors have also shown that this virus can transduce human lung ex vivo. Studies are sufficiently powered and described.

Comment #1:

First is that both ciliated epithelial cells and ATII cells appear to be targeted by AAV6.2FF. A more detailed analysis of what these ciliated cells are would be valuable. Further, only FACS data is provided to support the “cell-specificity” of the GFP virus. Immunofluorescent staining of thin sections from mice given AAV6.2FF-GFP showing distribution of delivery and GFP transgene expression would be of value. While they stain for SPB expression in SBP compound transgenic mice in following doxycycline withdrawal, staining is difficult to visualize and it appears that the number of SPB positive cells is much less than 21.5% of ATII cells as stated for AAV6.2FF-GFP FACS analysis.

We thank Reviewer #1 for their suggestions and comments, particularly in their focus on the transduction of AAV6.2FF into off target (non AT2 cell) cells and the implications this has on patient safety and clinical utility. We agree that we did not fully explore these questions in our original manuscript, however, we have carried out a series of extensive and detailed novel experiments to address these issues. To demonstrate the specificity of the AAV6.2FF vector targeting AT2 cells, we generated an AAV6.2FF-emGFP-nlsCre vector and administered this into Rosa 26 floxed-LacZ reporter mice (Rosa26-Flox/LacZ; JAX Stock No 003474) by intranasal (IN) administration (**Figure 1A-B**). Any cell transduced by AAV6.2FF will result in the Cre recombinase-mediated removal of the loxP sites subsequently causing β -galactosidase (β -gal) expression. This activity can be observed by X-gal staining. Note that we did not assess GFP expression due to its well-documented issues with autofluorescence in the lungs (Davis AS et al., Characterizing and Diminishing Autofluorescence in Formalin-fixed Paraffin-embedded Human Respiratory Tissue, J Histochem Cytochem, 62, 405-23, 2014). We observed that a number of cells in the nasal cavity, airways and in the distal lung were transduced by AAV6.2FF and positive for X-gal staining (**Figure 1C-D**). However, IN delivery did not result in vector transduction of cells belonging to a number of other organs and tissues in the body (**Figure 1E**). This suggests that delivery through the nasal cavity or trachea by IN, intratracheal (IT) or intubation delivery localizes transduction to airway or distal lung cell types. We address the potential effects that this transduction of multiple respiratory cell types has in Comment #2.

To respond to the second point from Reviewer #1 in Comment #1 on the need for visual evidence to support our flow cytometry results, we performed immunofluorescence (IF) staining in SP-B deficient mice administered AAV6.2FF–Luciferase (**Figure 2A**). We performed long-term (183 days) IF staining as we have already demonstrated short-term expression from the AAV vector by IF imaging (3 days to 60 days). We observe luciferase (Luc) expression by IVIS imaging 7 days after delivery, and verify expression 182 days after AAV6.2FF vector delivery (**Figure 2B**). The lungs from these mice were harvested 1 day following IVIS imaging on Day 182 and processed for IF staining. The IVIS images were confirmed by IF imaging, and we observe that Luc co-localizes with proSP-C in AAV-mSPB treated mice (**Figure 2C**). We also improved all IF images from the original manuscript submission (**Pages 7-11 of this document**).

With respect to the final comment, it has been difficult to demonstrate that 21.5% of cells are transduced by AAV6.2FF by IF staining as the level of SP-B expression often depends on which regions of the distal lungs AAV-SPB transduced. Choosing the correct field of view is critical and SP-B staining is inconsistent and heterogenous in different fields of view.

Figure 1.

Figure 1.

Figure 1. AAV6.2FF expressing Cre mediates functional recombination in multiple cell types in the mouse respiratory tract and distal lungs, but does not transduce other organs following intranasal administration. (A) Schematic of the recombinant AAV2 (rAAV2) vector genome containing a CMV promoter, emerald(em)-GFP, a nuclear localization signal (nls) and Cre recombinase (AAV6.2FF-emGFP-nlsCre) pseudotyped with the AAV6.2FF capsid. (B) Study design to demonstrate the extent of airway and lung cell transduction 3 weeks following modified intranasal (IN) instillation (Santry LA et al., AAV vector distribution in the mouse respiratory tract following four different methods of administration, BMC Biotech, 17:43, 2017) of phosphate-buffered saline (PBS) or 10^{11} vector genomes (vg) of AAV6.2FF-emGFP-nlsCre into 6-week-old Rosa26-Flox/LacZ (B6.129S4-Gt(ROSA)26Sor^{tm1Sor}/J) mice. (C) Whole-mount X-gal staining of the lungs, nasal cavity and trachea 3 weeks post-vector administration from both the PBS and AAV6.2FF-emGFP-nlsCre mice show that only the mice administered AAV were positive for β -Gal activity in the lungs, nasal cavity and the trachea. (D) Representative nuclear fast red counterstained paraffin sections (4 μ m) from X-gal stained lungs. Abundant LacZ-positive cells were found in the distal airway epithelial cells and alveolar cells of Rosa26-Flox/LacZ mouse lungs infected with AAV6.2FF-CMV-emGFP-nlsCre. Morphologic criteria demonstrate that both ATII (black arrow heads) and Club (Clara) cells (yellow arrow heads) are X-gal positive in the alveolar regions of the lung. LacZ-positive epithelial cells are also observed in tracheal and airway epithelium. (E) AAV6.2FF expressing Cre does not mediate functional recombination in all non-lung tissues (brain, heart, liver, kidneys, pancreas, spleen) examined following IN delivery. (**These figures are Figure 1f and Supplemental Figure 2a-e in the manuscript**)

Figure 2.

Figure 2. Co-staining to support flow cytometry data demonstrates long term co-expression of luciferase delivered by the AAV6.2FF vector in proSP-C positive AT2 cells. (A) Study design to visually support our flow cytometry results by immunofluorescence (IF) imaging. **(B)** IVIS imaging at 7 days and 182 days post AAV administration demonstrates rapid and long-term AAV expression of firefly luciferase (Luc). **(C)** IF imaging of Luc (green) and proSP-C (red) 183 days post AAV administration demonstrating long term (183 days) Luc expression in AT2 cells. Staining for proSP-C, Luc, and co-localization are indicated by the white arrows. **(These figures are Supplemental Figure 3g-l in the manuscript)**

To address the third point from Reviewer #1 in Comment #1 on the difficulty in visualizing the IF images in the manuscript, we improved the quality or increased the magnification of all IF images from the original manuscript submission. Reviewer #2 also commented on the poor quality of the IF images. The before and after improvements to the quality of the IF lung images are presented in the following pages and the improved figures are included in the revised manuscript submission.

Figure 2j.

Figure 2j.

Figure 4g.

Supplemental Figure 5d.

Supplemental Figure 5e.

Supplemental Figure 5e.

Supplemental Figure 5f.

Supplemental Figure 5g.

Supplemental Figure 6f.

Supplemental Figure 9h.

**Comment #2:**

A second concern also focuses on the non-ATII cells that are targeted by AAV6.2FF. If these cells are indeed targeted by the virus, what is the effect of SPB overexpression in cells that do not normally express SPB? How would this affect any translation to the clinical setting?

Based on (Figure 1) of these manuscript revisions as well as our original flow cytometry experiment, we now know that multiple cell types including AT2 cells are transduced by AAV6.2FF. How this affects translation to the clinical setting is currently unclear. However, all our pre-clinical experiments using the transgenic SP-B small animal model or human PCLS suggests that no adverse effects are caused by this vector. This conclusion is based on four lines of evidence. The first was body weight measurements. It is well recognized that body weight is a primary indicator of the overall well-being and health of the animal (Burkholder T et al., Health Evaluation of Experimental Laboratory Mice, Curr Protoc Mouse Biol, 2, 145-65, 2012), and has been used in previous small animal pre-clinical studies to demonstrate the safety and effectiveness of gene therapies such as Zolgensma (onasemnogene abeparvovec-xioi) (Foust KD et al., Rescue of the spinal muscular atrophy phenotype in a mouse model by early postnatal delivery of SMN, Nat Biotech, 28, 271-76, 2010). In our study, AAV6.2FF administration resulted in similar increases in body weight as untreated control mice (Manuscript Figure 1b). But the most striking example of the therapeutic effects of AAV-SPB is following dox removal. In control or untreated mice, dox removal results in rapid weight loss which is a clear indicator of impending lethal respiratory distress. These mice do not eat or drink as they simply try to focus on the act of respiration. This is in contrast to the behavior of AAV-SPB treated mice following dox removal which results in these mice displaying increases in body weight slightly less than age matched mice on dox. This is best demonstrated in our inflammatory and neonatal studies where we compare weights both before and after AAV administration to mice maintained on dox. In the original manuscript submission, a number of body weight measurements

demonstrated that the AAV6.2FF vector did not hinder increases in body weight. In a study treating SP-B transgenic mice with a high dose (5×10^{11} vg) of AAV-mouse(m)SPB, there was a regular increase in body weight for >100 days after AAV administration (Manuscript Figure 3c). In an experiment where SP-B mice were treated with an intermediate dose (10^{11} vg/mouse) of AAV-human(h)SPB, mice demonstrated regular weight gain for >100 days after AAV administration. In our manuscript revisions, we also assessed body weight in each of our studies but this time compared weight gain with mice on dox and showed comparable increases in weight in AAV-mSPB or AAV-hSPB treated mice in both neonates and adults (**Figures 4B, 5B, 5G, 6B, 6F**). The only time that we see AAV-SPB treated mice lose weight, is when the therapeutic effects of AAV-SPB start to wane.

The second line of evidence that AAV6.2FF is safe were the measurements of inflammatory cytokines following AAV administration. We looked for signs of systemic inflammation based on serum cytokine measurements and found that AAV-mSPB treatment lacked an inflammatory cytokine profile 1, 7, and 29 days after AAV administration (**Figure 6D-E**). Furthermore, there were no signs of inflammatory cells such as neutrophils present in neonatal mice 3 days after AAV-SPB treatment (**Figure 5I-K**).

The third line of evidence has been the long-term survival of mice following AAV-SPB treatment. From our survival studies we know that median survival was as high as 194 days in mice treated with an intermediate dose of AAV-mSPB, and up to 293 days in a single animal. This suggests no adverse effects were observed for more than 200 days following AAV-SPB treatment. Visual evidence demonstrates similar behavior and movements in AAV-SPB treated mice off dox, as untreated control mice on dox (**Figure 5L-M**).

The fourth line of evidence is our metabolic resazurin assay of precision cut lung slices (PCLS) from human embryos. Three days after transduction with 10^{10} vg/well of AAV-mSPB or AAV-hSPB these tissue slices show the same viability as measured by metabolic activity as untransduced PCLS (**Figure 8C**).

In conclusion, even though our results indicates that non-AT2 cells are transduced by AAV6.2FF, regular body weight gains, the lack of systemic inflammation or inflammatory cells, long term survival, and maintenance of human lung tissue viability suggests that AAV6.2FF targeting of non AT2 cells does not cause any obvious adverse events. However, we realize that off target transduction of AAV6.2FF will be of concern to regulatory bodies, and any advancements of this therapy towards clinical translation will be preceded by safety and expression studies conducted in large animal models. These are studies we are currently planning to conduct within this year.

We also looked at the effects of SP-B expression in other cell types in the literature. SP-B is not just expressed from AT2 cells but is also found in non-ciliated Club (Clara) cells in the pulmonary epithelium. Club cells are especially common in the conducting airways and terminal bronchioles but are not found in the alveoli (Lin S et al., SP-B^{-/-} Mice Are Rescued by Restoration of SP-B Expression in AT2 Cells But Not Clara cells, JBC, 274, 19168-174, 1999). Our experiments demonstrate multiple cells in the trachea and conducting airways are transduced by AAV6.2FF including Club cells (**Figure 1D**). In the study from 1999, when transgenic mice were generated to express SP-B from AT2 cells or Club cells in SP-B^{-/-} mice, only AT2 cell expression restored proper pulmonary surfactant homeostasis and respiratory function. When SP-B expression is restricted to Club cells, mature SP-B is unable to be generated as Club cells do not have the ability to process the SP-B proprotein, and these animals subsequently die of respiratory distress (Lin S et al., 1999). Thus AAV-SPB transduction of other cell types likely

results in increased expression of SP-B proprotein but not mature SP-B in cells. So far, all our studies indicate that the targeting of different respiratory cells does not appear to cause any noticeable adverse events.

Comment #3:

A third concern is one of cell-specificity of this virus. While the authors show no endothelial cell transduction in the lung following intratracheal delivery, this is not surprising given that endothelial cells are not available to the virus by this delivery route. It could be valuable to explore the “cell-specificity” of this virus when delivered by various routes (IV, IM, etc) or in cultured cells other than pulmonary epithelium.

We have previously demonstrated that AAV6.2FF rapidly (24 hr) transduces muscle tissue and for an extended period of time (at least 206 days) following intramuscular (IM) gastrocnemius administration (van Leishout et al., A Novel Triple-Mutant AAV6 Capsid Induces Rapid and Potent Transgene Expression in the Muscle and Respiratory Tract of Mice. *Molecular Therapy - Methods & Clinical Development* **9**, 323-329, 2018). To further explore Reviewer #1's question on cell specificity of AAV6.2FF, we intravenously (IV) administered AAV6.2FF-emGFP-nlsCre by tail vein injection into Rosa26-Flox/LacZ mice. Three weeks after IV injections, PBS or AAV6.2FF-emGFP-nlsCre mice were harvested and whole mount and histology of organs were performed. The extent of β -Gal activity in various tissues including the brain, lungs, heart, liver, kidneys, pancreas and spleen were assessed by X-gal staining. We observed recombination in the lungs, heart, liver and spleen but no X-gal staining in the brain, kidneys, or pancreas (**Figure 3**). This demonstrates that AAV6.2FF can target various tissues or cell types when delivered IV. This also suggests that if the vector is delivered through the nasal cavity or trachea (IN (**Figure 1**), intratracheal, intubation), non-respiratory tissues and organs do not undergo AAV6.2FF vector transduction.

Figure 3.

Figure 3. AAV6.2FF expressing Cre recombinase mediates functional recombination in the lung, heart, liver and spleen but not in the brain, kidneys or spleen following intravenous administration. Six-week-old Rosa26-Flox/LacZ (B6.129S4-Gt(ROSA)26Sor^{tm1Sor}/J) mice were administered phosphate-buffered saline (PBS) or 10^{11} vector genomes (vg) of AAV6.2FF-CMV-emGFP-nlsCre IV via tail vein injection. X-gal staining of tissues was performed 3 weeks post-vector administration, and both representative whole mount gross organ images, as well as nuclear fast red counterstained cryosections (8 μ m) from X-gal stained tissues are shown. (***This figure is Supplemental Figure 4 in the manuscript***)

Reviewer #2:

Kang et al. here describe the use of a lung tropic AAV vector to deliver a murine Surfactant Protein B (SP-B) construct to improve survival and surfactant function in a mouse model of SP-B deficiency. Using a CMV gene enhancer region, a chicken β -actin promoter and either myc-tagged SP-B construct or a luciferase reporter, the authors show that the AAV vector mediates long-term gene expression that appears to be specific to EpCAM positive lung epithelial cells with an efficiency of approximately 20%. The mouse model of SP-B deficiency is one where a doxycycline-driven SP-B construct is expressed in a SP-B deficient mouse. Thus, removal of doxycycline results in SP-B deficiency and fairly rapid death in the absence of treatment. Administration of AAV-SP-B a month before removal of doxycycline was associated with robust expression of SP-B in Ep-CAM+ cells in the absence of doxycycline and loss of SP-B transgene expression, with evidence of lung protection from acute injury and maintenance of lamellar bodies in alveolar type 2 (AT2) cells. Additionally, PV curves, deflation stability and compliance were improved in mouse lungs treated long-term with AAV-SP-B. Studies to determine dose-dependence and the time course for transgene expression showed that increasing dose of AAV-SP-B was associated with increased length of survival that was unaffected by administration of the vector with exogenous surfactant. Long-term AAV-SP-B expression prevented lung injury and surfactant dysfunction from SP-B deficiency. Finally, the authors demonstrate that this vector can transduce precision cut human lung slices from human fetal lungs. The manuscript is well written, extensive data with appropriate controls are presented, and the data are internally consistent. The following issues arise from review of this manuscript:

Comment #1:

While the model used to induce SP-B deficiency is established and results in profound SP-B deficiency, it does not mimic the human condition appropriately. Human infants with congenital SP-B deficiency are never exposed to SP-B and develop respiratory distress immediately after birth. Identification of these infants for gene therapy is difficult as the thought that this could be a genetic defect of surfactant is not high in the differential diagnoses in neonates with acute respiratory failure at birth. Further, the current studies are unable to address the effect of SP-B exposure in a previously naïve state with respect to the induction of anti-SP-B antibodies by the host. This could alter the efficiency of transgene expression and, more importantly, hamper future lung transplantation to correct the abnormality. The experiments conducted are also in adult and not in neonatal mice who have not yet completed full lung development, in particular alveolarization. The dose response to AAV-Luc and AAV-SP-B needs to be evaluated in neonatal mice with particular attention paid to the effects on inflammation and alveolarization. Further, in order to be translatable to the human condition, it is possible that in utero therapy would need to be achieved. Perhaps the use of mice heterozygous for the SP-B gene should be used for in utero or early neonatal treatment studies to validate this approach. Thus, the authors' contention that the studies described here will result in rapid translation to the human appears to be premature. Indeed, the model is described as one of inherited SP-B deficiency. This is not accurate as this model is a loss of SP-B and more akin to acquired SP-B deficiency. This should be clarified in the manuscript, especially the title.

The authors thank Reviewer #2 for their comments and suggestions on how to improve the accuracy and the clinical relevancy of this manuscript. We agree with much of their criticisms and suggestions and have conducted novel experiments to respond to their concerns. We answer Reviewer #2's queries on a point-by-point basis.

The first point we focus on is their concern with our study design and how gene therapy was administered before the development of respiratory distress and is therefore a treatment for acquired SP-B deficiency and not inherited SP-B deficiency. We agree with this point and subsequently designed a study in which dox was removed 2 days (-2D) and 1 day (-1D) before AAV administration (**Figure 4A**). Mice in the 1×PBS + BLES control group had dox removed 1 day before administration (-1D) and had a median survival of 3.65 days. In comparison, AAV-hSPB treatment has resulted in a median survival in both groups (-2D and -1D) which has not yet been reached (at 112 (-1D) and 113 (-2D) days) at the time of this manuscript revision submission. We were unsure whether SP-B would express rapidly enough as single stranded AAV requires second strand synthesis, however what is especially encouraging is the improvement in median survival between the 1×PBS + BLES treated mice and the -2D AAV-hSPB + BLES treated mice (**Figure 4C**). The -1D AAV-hSPB + BLES group had a low n number (n=6) that limited the power to detect differences in survival. This study also suggests that SPB expresses much more rapidly from the AAV6.2FF vector than initially thought as it would be necessary to express within 1 to 2 days in order to rescue the lethal respiratory distress in the -2D group which was without dox for 2 days before gene therapy.

The second point we address is with regards to the difficulties in the identification of SP-B patients due to the multiple potential diagnostic possibilities of neonates suffering acute respiratory distress. A previous study has stated that between 1993 (the first year that prospective diagnosis of SP-B infants became available) and 2005, 33 patients were identified with SP-B deficiency in North America. Diagnosis before evaluation for lung transplantation occurred for 32 out of 33 patients at a median age of 19 days of age, while 1 out of 33 was diagnosed after undergoing a lung transplantation (Palomar LM et al., Long-term Outcomes After Infant Lung Transplantation For Surfactant Protein B Deficiency Related To Other Causes Of Respiratory Failure, *J Pediatrics*, 2006). This suggests that diagnosis of SP-B deficiency can occur as early as ~19 days of age.

The third point we address is the effect of delivering AAV-SPB to an individual in a naïve state with respect to SP-B expression and the problems this may cause for future lung transplantation. Between 1993 and 2005, out of the 32 SP-B patients offered a lung transplantation, only 17 of the family's patients went ahead with the lung transplantation. And of these 17 patients, only 12 survived long enough to receive a lung transplantation. The 5-year survival rate of the 12 patients receiving the lung transplant was 48%. In some cases, especially in Europe, SP-B patients may not be offered the option of a lung transplantation. Therefore, AAV-SPB gene therapy may represent the only therapeutic option for some patients and their families. It is true that antibodies may develop in AAV-SPB treated patients, but this could even be caused by the administration of exogenous surfactant used to treat these neonates for their acute respiratory distress symptoms. In SP-B patients that received a lung transplant, 3 out of 7 developed circulating SP-B antibodies. The culprit that induced the antibody response, either exogenous surfactant or the lung transplantation, was unknown but fortunately none of these patients demonstrated any adverse reactions to the SP-B antibodies (Palomar LM et al., Long-term Outcomes After Infant Lung Transplantation For Surfactant Protein B Deficiency Related To Other Causes Of Respiratory Failure, *J Pediatrics*, 2006). There are two factors that may

mitigate the development of SP-B antibodies in neonatal patients receiving AAV-SPB. One, is that these patients are at an age (first 3 months of age) when their cellular immune system is still rapidly maturing (Basha S et al., Immune Responses in Neonates, Expert Rev Clin Immunol, 10, 1171-84, 2014), and this may limit their ability to develop antibodies. Two, infants receiving lung transplants will undergo a robust regimen of multiple drug immunosuppression (Huddleston CB et al., Lung Transplantation in Very Young Infants, J Thoracic & Cardiovasc Surgery, 118, 796-804, 1999) which may also hinder SP-B antibody development. In the SP-B animal model, we are currently working on the extent of SP-B antibody development following AAV-SPB administration and testing novel strategies to mitigate the immune response in order to readminister AAV-SPB. This is a unique problem that has little precedence, but any advancement of this therapy to the clinic by our group will be preceded by significant safety studies including the assessment of SP-B antibody development.

The fourth point we address is Reviewer #2's concern that adult mice and not neonatal mice were used in this study. To address this matter, we designed a study in which transgenic SP-B deficient mice at neonatal ages P8-P10 were either left on dox, treated with 1×PBS + BLES, or treated with 10¹¹ vg/mouse AAV-hSPB + BLES (**Figure 5A**). At P21, the mice were weaned off their dox diet onto a regular chow diet (except for the mice maintained on dox) and body weight and survival were assessed in all 3 groups. Neonatal mice treated with AAV-hSPB + BLES demonstrate similar increases in body weight as mice on dox (**Figure 5B**), and all mice (7 of 7) treated with AAV-hSPB remain alive (at 92 days post dox) at the time these revisions were submitted (**Figure 5C**). It should be noted that these surgeries were difficult as the volume of fluid instilled into the lungs of these P8-P10 pups resulted in ~50% mortality due to the inability of the pups to survive the injection volume (20µL) into their smaller lungs. At P8 7/10 (70%) died, at P9 4/11 (36%) died, at P10 2/5 (40%) died within minutes of administration, and death occurred regardless of whether 1xPBS or AAV-hSPB was instilled. To reduce post-surgery death, we repeated this experiment into older neonatal pups (P13-P15) which significantly reduced mortality (1/27 or 4%). To further enhance the clinical relevancy of this experiment, we removed the pups off dox immediately following AAV administration by placing them with FVB/N mothers that had given birth to similar aged pups (**Figure 5F**). This required setting up equivalent breeding pairs of transgenic SP-B mice (6 pairs) and FVB/N mice (6 pairs). FVB/N mice is the background strain that the SP-B transgenic mice are derived from. As with the first study using neonatal pups, AAV-hSPB treatment resulted in similar body weight increases as mice on dox (**Figure 5G**), and a significant improvement in median survival compared to 1×PBS + BLES treated mice (**Figure 5H**). At the time of manuscript revision submission there were still 6 out of 8 mice alive at 68 days post dox removal.

The fifth point we address is whether AAV administration results in inflammation or affects alveolarization. In our second study using neonatal mice, we removed 2 mice from each of the 3 groups (**Figure 5F**) and performed H&E staining to look at alveolar structure (**Figure 5J**), and WGJ to look for the presence of inflammatory cells such as neutrophils (**Figure 5K**). Lung histology 3 days after AAV administration + dox removal revealed normal alveolar structure, and no signs of inflammatory cells such as neutrophils were observed (**Figure 5**).

Finally, as suggested by the reviewer, we are currently testing the use of our AAV6.2FF vector in utero for future studies. In particular, using the AAV6.2FF vector to deliver gene editing constructs in the mouse model of SP-B deficiency.

Figure 4.

Figure 4. Demonstration that AAV-SPB is a therapeutic strategy for inherited SP-B deficiency. (A) Study design demonstrating whether AAV-hSPB therapy is a feasible treatment for inherited SP-B deficiency. (B) Gene therapy restored long-term body weight gain in both AAV-hSPB treatment groups, whereas 1xPBS + BLES treated mice rapidly suffered weight loss before undergoing lethal respiratory distress. (C) 10¹¹ vg/mouse of AAV-hSPB + BLES therapy improved median survival in both groups to an undefined number of days compared to a median survival of 3.65 days in 1xPBS + BLES treated mice. At the time of manuscript revisions, median survival in the -D1 group was at least \geq 110 days, and in the -D2 group was at least \geq 111 days. There are still 4 of 6 mice alive in the -D1 group and 6 of 8 mice alive in the -D2 group. (D) A video of mice in the -D2 group 111 days after dox removal demonstrates no signs of respiratory distress. (These figures are Figure 5a-c in the manuscript)

Figure 5.

Figure 5.

Figure 5. AAV-SPB ameliorates lethal respiratory distress in an inherited neonatal mouse model of SP-B deficiency. (A) Study design to assess the therapeutic efficacy of AAV-hSPB treatment in neonatal mice. (B) AAV-hSPB gene therapy resulted in comparable increases in body weight as mice on dox, whereas the 1×PBS + BLES treated mice developed an impairment in weight gain before rapidly undergoing lethal respiratory distress following dox removal. (C) 10^{11} vg/mouse of AAV-hSPB + BLES therapy improved median survival in neonatal mice to an undefined number of days compared to a median survival of 4.94 days in 1×PBS + BLES treated neonatal mice. At the time of submission of manuscript revisions, median survival was at least ≥ 90 days with all AAV-hSPB treated mice (7 out of 7) remaining alive. (D) Images of the intratracheal injection into a P8 pup, as well as P8 and P10 injected pups 5 hr and 24 hr post surgery respectively. (E) Video of mice treated when they were P10 neonatal pups 90 days after weaning (dox removal). (F) We repeated AAV-hSPB administration into older pups (P13-P15) and studied an inherited model of SP-B deficiency by removing them off dox immediately following AAV administration by placing them with FVB/N mothers that had given birth to similar aged pups. (G) As in **Fig B**, gene therapy resulted in equivalent increases in body weight to mice on dox, whereas 1×PBS + BLES treated mice displayed an impediment in weight gain before undergoing rapid lethal respiratory distress. (H) 10^{11} vg/mouse of AAV-hSPB + BLES therapy improved median survival in neonatal mice to an undefined number of days compared to a median survival of 5.98 days in 1×PBS + BLES treated neonatal mice. At the time of manuscript revision submission, median survival was at least ≥ 66 days with 6 out of 8 mice still alive. (I) To assess whether AAV-hSPB had effects on lung structure and alveolarization in neonatal mice, lungs (n=2 per group) were harvested for H&E and WGJ staining from neonatal mice 3 days after administration of AAV-hSPB + BLES plus dox removal. Mice treated with 1×PBS + BLES show some signs of lung damage (yellow arrows). (J) Histological images (H&E and WGJ) of the lungs from AAV-hSPB mice show normal alveolar structure, whereas lungs from 1×PBS mice demonstrate either infiltration by red blood cells (ID: 1×PBS, B) or septal wall thickening (ID: 1×PBS, A). There are no signs of inflammatory cells such as neutrophils in the AAV-hSPB treated mice. (K) WGJ staining of a case of human appendicitis provided by the University of Ottawa Histology Core Facility shows the presence of numerous inflammatory cells such as multilobed

neutrophils (red arrows) (Matute-Bello, G., et al., and the Acute Lung Injury in Animals Study Group. An official American Thoracic Society workshop report: features and measurements of experimental acute lung injury in animals. *Am J Respir Cell Mol Biol* 44, 725-738, 2011). (L) Video from mice on dox at 81 days of age demonstrating their size and activity levels. (M) Video from neonatal mice 66 days after AAV-hSPB + BLES treatment and dox removal at 80 days (n=2) and 82 days (n=2) of age showing similar activity levels and body weight as the control mice on dox in **Fig L**. (***These figures are Figure 4l-n, Supplemental Figure 9p-r, Figure 5d-h, and Supplemental Figure 10a-b in the manuscript***)

Comment #2:

The effects on inflammation have not been addressed in any of the experimental conditions examined. This should be included in the manuscript for the long-term and day 3 and 7 experiments.

To address the effects of AAV-mSPB on the development of systemic inflammation we repeated the long-term day 7 experiment (Manuscript Figure 3d). We compared the effects of AAV administration to lipopolysaccharide (LPS), a highly endotoxic component of the Gram-negative bacteria outer membrane. Administration of LPS into animals elicits a wide physiological response and we measured body weight, survival and inflammatory cytokine levels. LPS induced systemic inflammation results in the highest cytokine levels 2 to 12 hr after injection, with levels returning to baseline within 12 to 24 hr (Seeman S et al., *Comprehensive comparison of three different animal models for systemic inflammation*, *J Biomed Sci*, 24:60, 2017). In this experiment mice were treated with: 1×PBS (n=7) by IT administration; 10¹¹ vg AAV-mSPB (n=8) by IT administration; and 3mg/kg of LPS (Sigma) (n=8) by a single bolus intraperitoneal (IP) injection. The 1×PBS mice were maintained on dox to allow for the comparison of cytokine levels to the AAV-mSPB treated mice over approximately 30 days. The dosage of LPS we used is considered sublethal in adult mice, however 5 of 8 mice died post IP LPS injection. Serum was collected 7 days before (-7D) and 1 day (+1D), 7 days (+7D), and 29 days (+29D) after AAV or LPS administration. The dox was removed from the AAV-mSPB treated group 7 days after vector delivery on the day of the -7D serum collection (**Figure 6A**). Thirteen humoral factors were analysed by multiplex technology using the LEGENDplex™ (BioLegend) mouse inflammation panel. This assay is a bead-based immunoassay operating on a similar principle to the sandwich immunoassay. The samples were analysed in a V-bottom plate on a BD FACSCanto II flow cytometer using a high throughput autosampler. From -D7 to +D1, LPS still demonstrated a significant increase in inflammatory cytokines 24 hr after injection with IL-23 ($P=0.0203$), TNF α ($P=0.0362$), and GM-CSF ($P=0.0456$), and a non-significant trend towards an increase in cytokine levels with IL-6 and IFN- γ . AAV-mSPB administration did not result in a significant increase from -D7 to +D1 in any of the 13 cytokines measured (**Figure 6D-E**). All statistics were performed using the Mixed Effects Model with the Geisser-Greenhouse correction followed by a post hoc Tukey's multiple comparison's test. Note that data was analyzed by fitting the Mixed Model rather than by repeated measures ANOVA as there were missing values due to the death of animals (5 LPS and 1 AAV-mSPB mice) during the course of the study.

Figure 6.

Figure 6. AAV-mSPB administration does not cause systemic inflammation as measured by humoral factors (cytokines). (A) Serum collected from the lateral saphenous vein of SP-B deficient mice in a survival study 7 days before (-7D), and 1 (+1D), 7 (+7D), and 29 days (+29D) after AAV or LPS administration, were analysed for 13 mouse cytokines using the LEGENDPlex bead-based immunoassay mouse inflammation panel. (B) The negative control group was administered with 1xPBS and maintained on dox, while a control group for increased inflammatory cytokines was administered with 3mg/kg of lipopolysaccharide (LPS) by intraperitoneal (IP) injection. Body weight in AAV-mSPB treated mice is comparable to the on dox control group even when dox is removed, while LPS treated mice rapidly developed weight loss before stabilizing and recovering 72-96 hr after LPS injection. (C) Median survival until the final serum collection. (D) Measurements of 13 mouse inflammatory cytokines from the 3 groups before (-7D) and after (+1D, +7D, +29D) AAV or LPS administration. (E) Individual cytokine measurements from the 3 groups over the 4 different time-points. (F) Body weight over time demonstrates that mice administered AAV-mSPB maintain similar values as mice on dox. (G) Survival for the entire duration of the study. At the time of manuscript revision submission, median survival was at least ≥ 109 days with 7 of 8 AAV-mSPB treated mice still alive in this study. There is no significant difference in median survival between mice on dox and AAV-mSPB treated mice, while mice injected with LPS had a median survival of 1.95 days with only 3 out of 8 mice remaining alive as of manuscript revision submission. (These figures are Figure 3e-h and Supplemental Figure 8f-g in the manuscript)

Comment #3

The authors show Ep-CAM⁺ cells being preferentially transfected with AAV-SP-B. However, the efficiency of transfection is only demonstrated for the AAV-Luc construct. The authors should demonstrate the efficiency of AAV-SP-B transfection by determining the number of SP-B positive cells as a proportion of total Epi-CAM⁺ cells.

Here, we present visual evidence of AAV-SPB transduction of EpCAM positive cells by co-staining SP-B with EpCAM in the lungs of mice that have been administered with AAV-mSPB. These lung samples are from Figure 2j of the original manuscript submission. It was difficult to determine the percentage of SP-B positive cells compared to total EpCAM⁺ cells using flow cytometry due to the lack of appropriate antibodies as SP-B is a cytosolic soluble protein.

Figure 7.

A.

B.

C.

Figure 7. Representative epifluorescence images of EpCAM (green), SP-B (red), and DAPI (blue) from frozen lung sections of: **(A)** a mouse on dox; **(B)** an untreated mouse off dox for 9 days; and **(C)** a 10^{11} vg AAV-mSPB treated mouse off dox for 60 days. Only the mice on dox or treated with AAV-mSPB demonstrate co-localization of SP-B with EpCAM.

Comment #4

The authors contend that the administration and effects of AAV-SP-B were enhanced by using exogenous surfactant delivery (BLES). However, Fig.3d and Suppl Fig 6 do not support this conclusion.

Reviewer #2 is correct and the addition of BLES as a vehicle does not statistically improve AAV distribution or survival in healthy animals. This has been corrected in the manuscript and is highlighted in yellow.

Comment #5:

Interestingly, the effects of AAV-SP-B on survival are better in a model in which doxycycline is removed three or seven days after AAV-SP-B administration. This could be defined as the time to 50% mortality which appears to be 125 days in the model where AAV-SP-B is administered a month before doxycycline withdrawal and 150 days in the model where the transgene is administered 3 or 7 days before doxycycline withdrawal. Could this indicate a long-term adverse effect of the AAV vector?

We do not believe there is any long-term adverse effects with either the AAV6.2FF vector or the SP-B cDNA transgene. With respect to the AAV6.2FF vector, our long-term

luciferase/IVIS study showed that mice administered AAV6.2FF-Luc survived to at least 210 days before being harvested, and even outlived age-matched untreated control mice which had 2 mice dying before this 210-day timepoint. The AAV-Luc treated mice showed regular and consistent increases in body weight comparable to untreated mice.

We also did not observe adverse effects following AAV-mSPB or AAV-hSPB administration. The median survival for high dose AAV-mSPB treated mice that were on dox for 28 days after AAV administration was 128 days. This means they were exposed to AAV-mSPB for a total of 28 + 128 days = 156 days. The mice treated with an intermediate dose of AAV-mSPB in which dox was removed 7 days after AAV administration had a median survival of 194 days. This means they were exposed to AAV-mSPB for a total of 7 + 194 days = 201 days. The longest surviving mouse in this particular experiment survived for 287 days after dox removal which is 294 (287 + 7) days after AAV-mSPB administration. That means AAV-SPB did not result in any adverse effects in that one mouse for up to 294 days after administration.

However, it does appear that there are better results with AAV-SPB when dox is removed earlier. This is partly due to recording the days off dox when SP-B expression is high enough which is within days of administration but may also be due to overexpression of SP-B causing problems with surfactant protein homeostasis. It is possible that if there is too much SP-B, the AT2 cells transduced with the vector turnover quicker. When dox is removed at 3 or 7 days, the loss of endogenous SP-B combined with increasing SP-B expression from AAV-SPB maintains a more physiological surfactant concentration. However, as only SP-B deficient patients will receive this treatment, this should not be an issue as these patients will not be expressing any SP-B protein.

Comment #6:

Human lung precision cut slices were treated with AAV-Luc or AAV-mCherry constructs. The effects of the AAV-SP-B construct should also be shown to demonstrate the cell types that are transduced by such treatment in human tissues. Can an alveolar type 2 phenotype be driven by expression of SP-B in fetal human lung under these conditions?

We acquired new human fetal lung samples and generated PCLS (**Figure 8A-B**). These new lung slices were transduced with 10^{10} vg AAV-mSPB or 10^{10} vg AAV-hSPB. The AAV vectors had no adverse effects on the viability or metabolic activity of the lung slices 3 days after transduction (**Figure 8C**) demonstrating their safety. Untransduced PCLS shows some SP-B staining (**Figure 8D**). However, following transduction, the PCLS demonstrated robust increases of SP-B expression following AAV-mSPB (**Figure 8E**) or AAV-hSPB (**Figure 8F**) transduction.

Figure 8. PCLS from human fetal lungs demonstrate increased SP-B expression following AAV-mSPB or AAV-hSPB transduction. (A) Study design to assess expression of SP-B following AAV-SPB transduction of human PCLS. **(B)** Images of human PCLS in a 24-well TC plate 7 days after culture. A magnified image of a PCLS in an individual well as indicated by the white box. **(C)** A metabolic resazurin assay demonstrates equivalent cell viability between untransduced and AAV-SPB treated PCLS. **(D)** Untransduced human PCLS demonstrates minor SP-B expression. **(E)**

Human PCLS transduced with 10^{10} vg/mouse AAV-mSPB demonstrates robust SP-B expression. **(F)** Human PCLS transduced with 10^{10} vg/mouse AAV-hSPB demonstrates robust SP-B expression. **(These figures are Figure 6e-g in the manuscript)**

Comment #7:

In Suppl Fig 5c, why was 5×10^{11} vg/mouse used in these experiments as this was the optimal dose in Fig 3?

(Note that Supplemental Fig 5c has become Supplemental Fig 6c in the revised manuscript, however it is exactly the same figure as in the original submission) In Supplemental Figure 6c, it was actually an intermediate dose of 10^{11} vg/mouse that was administered to these mice. Supplemental Fig 6c was performed chronologically before the high dose (5×10^{11} vg/mouse) was used in Figure 3, and at the time we did not realize the highest dose was the optimal dose.

Comment #8:

The quality of the immunofluorescence and gross anatomy figures is poor. It is almost impossible to see the immunofluorescence and the gross pathology of the lungs are dark in all figures. In some images, the DAPI staining is also difficult to see (e.g. Suppl Fig 3). All these figures need to be improved substantially.

We have made improvements to both the gross anatomical lung and immunofluorescence (IF) images. Reviewer #1 brought up similar concerns regarding the poor quality of some of the IF images. Please see the answer to Reviewer #1 Comment #1 for improved IF images (**Pages 7-11 of this document**). The before and after improvements to the gross anatomical lung images are presented in the following pages and in the new manuscript figures (**Pages 30-32 of this document**).

Figure 2f.

Figure 2i.

Figure 4e.

Supplemental Figure 5i.

Supplemental Figure 6d.

Supplemental Figure 9e.

Comment #9:

The experimental methods are sufficiently detailed and appropriately catalogued.

All new experimental methodologies are included in the methods section, while all samples sizes and treatment variables are included in the supplemental figures.

Comment #10:

The statistical methods are appropriate.

For all new experiments, the statistical methods used are stated either in the body of the manuscript, the figure legends, or the methods section.

Reviewers' Comments:

Reviewer #1:

Remarks to the Author:

The authors have adequately addressed all concerns raised by the reviewers. This is a significant piece of work that will be of use to many in the field of pulmonary gene delivery for a variety of diseases.

Reviewer #2:

Remarks to the Author:

Kang et al. have responded well to the critiques of their manuscript, which is now significantly improved. The short-term removal of doxycycline experiments are particularly important. However, there are some issues that remain to be addressed.

1. The authors suggest that flow cytometry cannot be done for SP-B and EpCAM. However, have the authors tried fixing and permeabilizing the cells to allow intracellular access to the antibody? This would allow more accurate definition of the efficiency of SP-B expression.
2. The discussion should include the limitations of the models that the authors have used. Thus, even in the short term removal of doxycycline experiments, the mice are expressing SP-B, which is not the case in inherited SP-B deficiency.
3. Page 10, Lines 233 and 234: "a non-significant trend towards an increase...." This should be removed since the changes in IL6 and IFN-g are not significant.

The authors appreciate the time from both reviewers to read our manuscript and provide us with their comments and suggestions. We feel that their comments and experimental suggestions were crucial in improving the quality of this manuscript. Below is our point-by-point response to both reviewers' comments to our revised manuscript submission. The reviewers' comments are in bold black font, and our answers are in regular black font.

REVIEWERS' COMMENTS:

Reviewer #1 (Remarks to the Author):

The authors have adequately addressed all concerns raised by the reviewers. This is a significant piece of work that will be of use to many in the field of pulmonary gene delivery for a variety of diseases.

We again want to thank Reviewer #1 for their focus on the effects of non-AT2 cell transduction by AAV6.2FF. Their suggested experiments gave us new insights into the cell specificity of AAV6.2FF, and are directing our attention on identifying any potential adverse effects of this gene therapy vector. These experiments consist a large proportion of our future studies.

Reviewer #2 (Remarks to the Author):

Kang et al. have responded well to the critiques of their manuscript, which is now significantly improved. The short-term removal of doxycycline experiments are particularly important. However, there are some issues that remain to be addressed.

We thank Reviewer #2 for their many experimental suggestions including designing a study to test gene therapy against inherited SP-B deficiency, determining the effects of gene therapy in neonatal mice, and examining the inflammatory effects of AAV-SPB. These experiments improved the clinical relevancy of AAV-SPB gene therapy. We respond to Reviewer #2's most recent comments on a point-by-point basis.

1. The authors suggest that flow cytometry cannot be done for SP-B and EpCAM. However, have the authors tried fixing and permeabilizing the cells to allow intracellular access to the antibody? This would allow more accurate definition of the efficiency of SP-B expression.

We thank Reviewer #2 for this suggestion. This is a comment we addressed directly to the editorial staff at Nature Communications. Briefly, we stated that this methodological suggestion for fixing and permeabilizing was helpful, and this proposed experiment could give us novel quantitative data of SP-B levels that are expressed in EpCAM++ cells. Specifically, it could give us an idea of the percentage of EpCAM++ cells that are SP-B positive. However, we already performed this flow cytometry study using AAV-Luc, and want to emphasize that transduction efficiency and tropism is dependent on the capsid coat and not the transgene cargo. This quantitative value is also likely influenced by delivery methods as well as AAV-SPB dosage, and

the antibodies against SP-B in a flow cytometry experiment could not distinguish between endogenous SP-B and the exogenous SP-B delivered by AAV6.2FF. We have also visually demonstrated in a number of different experiments that AAV6.2FF efficiently transduces EpCAM++ or AT2 cells regardless of the transgene delivered by this vector.

2. The discussion should include the limitations of the models that the authors have used. Thus, even in the short term removal of doxycycline experiments, the mice are expressing SP-B, which is not the case in inherited SP-B deficiency.

We have included the following sentences into the Discussion section of the manuscript on page 16:

‘Although our gene therapy shows promising results, the study designs and mouse model still do not faithfully recapitulate the disease progression of human SP-B deficient patients that are born without SP-B protein expression. Even in this experiment where dox was removed before AAV administration (Figs. 9a-c), there was likely endogenous SP-B present when AAV was administered as suggested by SP-B staining in IF images of lungs from AAV-Luc negative control mice 3 days off dox (Supplementary Fig. 5d).’

3. Page 10, Lines 233 and 234: "a non-significant trend towards an increase...." This should be removed since the changes in IL6 and IFN-g are not significant.

This line has been removed from the manuscript.